# MissScore: High-Order Score Estimation in the Presence of Missing Data

**Wenqin Liu** [1]   **Haoze Hou** [2]   **Erdun Gao** [3]   **Biwei Huang** [4]   **Qiuhong Ke** [5]   **Howard Bondell** [1]   **Mingming Gong** [1][6]

## Abstract

Score-based generative models are essential in various machine learning applications, with strong capabilities in generation quality. In particular, high-order derivatives (scores) of data density offer deep insights into data distributions, building on the proven effectiveness of first-order scores for modeling and generating synthetic data, unlocking new possibilities for applications. However, learning them typically requires complete data, which is often unavailable in domains such as healthcare and finance due to data corruption, acquisition constraints, or incomplete records. To tackle this challenge, we introduce MissScore, a novel framework for estimating high-order scores in the presence of missing data. We derive objective functions for estimating high-order scores under different missing data mechanisms and propose a new algorithm specifically designed to handle missing data effectively. Our empirical results demonstrate that MissScore accurately and efficiently learns the high-order scores from incomplete data and generates high-quality samples, resulting in strong performance across a range of downstream tasks.

## 1. Introduction

The first-order derivative of the log data density, also known as (Stein) score (Liu et al., 2016), plays an important role in various machine learning applications, including data synthesis (Song & Ermon, 2019; 2020; Kim et al., 2022), super-resolution (Li et al., 2022), and inverse problems in

medical imaging (Song et al., 2021; Chung & Ye, 2022). Denoising Score Matching (DSM) (Vincent, 2011), an efficient method for estimating the score of the data density from samples, has become widely used in training score-based generative models (Ho et al., 2020; Song & Ermon, 2020). Beyond the first-order score, high-order derivatives of the data density, which we refer to as high-order scores, offer more refined local approximations of the data distribution, such as its curvature, and enable new model capabilities. For instance, they can improve the mixing speed of sampling methods and provide insights into quantifying the uncertainty in denoising problem (Dalalyan & Karagulyan, 2019; Sabanis & Zhang, 2019; Meng et al., 2021). Additionally, Lu et al. (2022) empirically demonstrated that incorporating high-order score matching improves the likelihood of score-based diffusion ordinary differential equations on both synthetic and real data, while maintaining high-quality generation. Furthermore, high-order scores have been utilized in recovering causal structures (Rolland et al., 2022; Sanchez et al., 2022; Liu et al., 2024).

Despite their promise, learning high-order scores usually requires training the model on complete data (Meng et al., 2021; Lu et al., 2022). However, in many real-world scenarios, such as healthcare, finance, and social networks, data often contain missing values due to privacy constraints or high sampling costs (Rubin, 1976; Shpitser, 2016). A straightforward approach to handling missing data is to impute the missing values and train the model on the imputed dataset. However, imputation methods often compromise data quality, and potentially leading to biased results and significantly degrading the performance of downstream tasks (Ouyang et al., 2023). Furthermore, these methods fail to capture the inherent uncertainty associated with missing data, producing a distribution of imputed values that poorly represents the true underlying data distribution (Van Buuren & Groothuis-Oudshoorn, 2011; Gondara & Wang, 2018; Wang et al., 2021). Some alternative approaches, such as using generative adversarial networks (GANs) or variational auto-encoders (VAEs) to directly approximate the data generation model from incomplete data (Li et al., 2019; Gain & Shpitser, 2018), require training additional networks, which can be computationally expensive and may also result in model inconsistency. To address this issue, some works propose to explicitly constrain the learning objective

---

[1]School of Mathematics and Statistics, The University of Melbourne, Australia [2]Institute of Statistics and Big Data, Renmin University of China, China [3]Australian Institute for Machine Learning, The University of Adelaide, Australia [4]Halıcıoğlu Data Science Institute, UC San Diego, United States [5]Faculty of Information Technology, Monash University, Australia [6]Department of Machine Learning, Mohamed bin Zayed University of Artificial Intelligence, United Arab Emirates. Correspondence to: Mingming Gong <mingming.gong@unimelb.edu.au>.

*Proceedings of the 42nd International Conference on Machine Learning*, Vancouver, Canada. PMLR 267, 2025. Copyright 2025 by the author(s).

within the model, which can enhance performance (Städler & Bühlmann, 2012; Gao et al., 2022). These studies emphasize the need for approaches that can directly and more efficiently handle missing data.

Back to score estimation from incomplete data, Ouyang et al. (2023) adopts a similar way by introducing a diffusion-based framework that learns the first-order score directly from incomplete data. In principle, high-order scores could be estimated from a learned first-order score model trained on incomplete data using automatic differentiation. However, this approach becomes computationally impractical for high-dimensional data and large model sizes, particularly when using deep neural networks based models (Meng et al., 2021). Furthermore, automatic differentiation introduces additional estimation errors, as small errors does not always lead to a small estimation error for high-order scores. Moreover, methods based on GANs (Goodfellow et al., 2020) or VAEs (Kingma, 2013) do not inherently capture score information, regardless of whether data is missing or complete. In contrast, score-based models naturally integrate this information (Li et al., 2019; Ho et al., 2020; Gain & Shpitser, 2018). These limitations highlight the need of using score-based models when estimating high-order scores in the presence of missing data.

**Contributions.** In this work, we introduce MissScore, a novel score-based framework for learning high-order scores in the presence of missing data. We derive objective functions for estimating these scores under different missing data mechanisms, using DSM to recover the true score function. While our framework is general and applicable to scores of any order, we focus on second-order scores (the Hessian of the log density) in our experiments. Our results show that MissScore efficiently and accurately approximates high-order scores with missing data. In addition, we demonstrate that our model improves both sampling speed and data quality in data generation tasks, with the quality of the generated samples validated across several downstream tasks.

## 2. Related Work

**Missing Data Problem.** Learning from incomplete observations is a common challenge in real-world datasets, arising from factors such as data corruption or incomplete records (Rubin, 1976; Little & Rubin, 2019). This issue has been extensively studied, with imputation being a primary solution (Poulos & Valle, 2018; Jäger et al., 2021; Shadbahr et al., 2023; Paterakis et al., 2024). Traditional imputation methods, such as mean or median substitution, tend to compromise data diversity and introduce biases in downstream tasks. More advanced techniques, including machine learning models and deep generative approaches like GAIN, MICE, and MIDA (Muzellec et al., 2020; Van Buuren & Groothuis-Oudshoorn, 2011; Gondara & Wang, 2018), aim

to capture data distributions more effectively. However, these methods are limited by their inability to model the inherent uncertainty in missing data and often struggle to outperform traditional methods in survey data (Wang et al., 2021).

Recent advances in generative modeling have explored learning directly from incomplete data, moving beyond explicit imputations. Variational autoencoder-based approaches handle complex tabular data through tailored likelihoods, hierarchical structures, or importance weighting (Nazabal et al., 2020; Mattei & Frellsen, 2019; Ma et al., 2020; Peis et al., 2022). Diffusion-based models offer an alternative by learning score functions over partially observed data, supporting both data synthesis and imputation (Ouyang et al., 2023; Zhang et al., 2024; Chen et al., 2024; Zheng & Charoenphakdee, 2022; Kotelnikov et al., 2023). However, many of these methods rely on auxiliary networks, make strong assumptions about the missingness mechanism, or are restricted to first-order scores. In contrast, our approach directly estimates high-order score functions from incomplete data, enabling richer local information and broader applicability.

**Score Matching.** Score matching estimates the gradient of the log-density, known as the score function, of a data distribution (Hyvärinen & Dayan, 2005). It is particularly effective for complex, high-dimensional density models with intractable partition functions. Among its variants, DSM (Vincent, 2011) has emerged as a foundational method, learning the score of a perturbed distribution by minimizing a regression loss. While first-order scores are widely used across applications (Song & Ermon, 2019; Li et al., 2022), higher-order scores offer richer insights by capturing the curvature of the data distribution (Dalalyan & Karagulyan, 2019; Sabanis & Zhang, 2019). These high-order derivatives improve sampling efficiency and uncertainty quantification (Meng et al., 2021). However, learning high-order scores is computationally expensive and prone to inaccuracies, particularly in high-dimensional settings. Recent work by Meng et al. (2021) addresses this challenge by leveraging Tweedie's formula to estimate high-order derivatives efficiently. Lu et al. (2022) further demonstrated that high-order score matching improves the likelihood of diffusion-based models while maintaining high sample quality. MissScore builds on these insights, extending DSM to high-order score estimation in the presence of missing data, ensuring both computational efficiency and robustness.

## 3. Estimating High-Order Scores with Missing Data

In this section, we provide an overview of high-order DSM and introduce our approach for handling missing data.

## 3.1. Background on High-Order DSM

Consider a data distribution $p_{\text{data}}(\mathbf{x})$ and a model distribution $p(\mathbf{x}; \boldsymbol{\theta})$ over $\mathbb{R}^d$. The first order score is the gradient of $\log p_{\text{data}}(\mathbf{x})$ with respect to $\mathbf{x}$, denoted as $\mathbf{s}_1(\mathbf{x}) = \nabla_{\mathbf{x}} \log p_{\text{data}}(\mathbf{x})$. Correspondingly, the score function of $p(\mathbf{x}; \boldsymbol{\theta})$ is denoted as $\mathbf{s}_1(\mathbf{x}; \boldsymbol{\theta}) = \nabla_{\mathbf{x}} \log p(\mathbf{x}; \boldsymbol{\theta})$. In DSM, instead of directly estimating the score function from the original data, the method works by introducing noise from a predefined noise distribution $q_\sigma(\tilde{\mathbf{x}}|\mathbf{x})$ into the data. The objective is then to estimate the score of the perturbed data distribution $q_\sigma(\tilde{\mathbf{x}}) = \int q_\sigma(\tilde{\mathbf{x}}|\mathbf{x}) p_{\text{data}}(\mathbf{x}) d\mathbf{x}$. To achieve so, DSM minimizes the following objective function,

$$\mathcal{L}_{\text{DSM}}(\boldsymbol{\theta}) = \frac{1}{2} \mathbb{E}_{p_{\text{data}}(\mathbf{x})} \mathbb{E}_{q_\sigma(\tilde{\mathbf{x}}|\mathbf{x})} \Big[ \| \mathbf{s}_1(\tilde{\mathbf{x}}; \boldsymbol{\theta}) $$
$$- \nabla_{\tilde{\mathbf{x}}} \log q_\sigma(\tilde{\mathbf{x}}|\mathbf{x}) \|_2^2 \Big]. \quad (1)$$

It has been shown that minimizing Eq. (1) is equivalent to minimizing the score matching loss between $\mathbf{s}_1(\tilde{\mathbf{x}}; \boldsymbol{\theta})$ and $\mathbf{s}_1(\tilde{\mathbf{x}})$ under certain regularity conditions (Vincent, 2011). When the noise distribution $q_\sigma(\tilde{\mathbf{x}}|\mathbf{x})$ is Gaussian, i.e., $\mathcal{N}(\tilde{\mathbf{x}}|\mathbf{x}, \sigma^2 \mathbf{I})$, the objective simplifies to

$$\mathcal{L}_{\text{DSM}}(\boldsymbol{\theta}) = \frac{1}{2} \mathbb{E}_{p_{\text{data}}(\mathbf{x})} \mathbb{E}_{q_\sigma(\tilde{\mathbf{x}}|\mathbf{x})} \left[ \left\| \mathbf{s}_1(\tilde{\mathbf{x}}; \boldsymbol{\theta}) + \frac{1}{\sigma^2} (\tilde{\mathbf{x}} - \mathbf{x}) \right\|_2^2 \right]. \quad (2)$$

The learned score function implicitly learns how to "denoise" the perturbed data $\tilde{\mathbf{x}}$, guiding it back toward the true data distribution through the optimization of Eq. (2). By focusing on estimating the score of the noise-perturbed distribution $q_\sigma(\tilde{\mathbf{x}})$ instead of the original data distribution $p_{\text{data}}(\mathbf{x})$, DSM offers a more efficient approach to score estimation compared to other techniques (Hyvärinen & Dayan, 2005; Song et al., 2020). Meng et al. (2021) provide a derivation of DSM using Tweedie's formula (Efron, 2011), and they generalize this approach to incorporate high-order moments of $\mathbf{x}$ based on $\tilde{\mathbf{x}}$, allowing them to develop an objective function for learning high-order scores.

**Theorem 3.1.** *(Meng et al., 2021)* $\mathbb{E}[\otimes^n \mathbf{x}|\tilde{\mathbf{x}}] = f_n(\tilde{\mathbf{x}}, \mathbf{s}_1, ..., \mathbf{s}_n)$, *where* $\otimes^n \mathbf{x} \in \mathbb{R}^{D^n}$ *denotes n-fold tensor multiplications,* $f_n(\tilde{\mathbf{x}}, \mathbf{s}_1(\tilde{\mathbf{x}}), ..., \mathbf{s}_n(\tilde{\mathbf{x}}))$ *is a polynomial of* $\tilde{\mathbf{x}}$, $\mathbf{s}_1(\tilde{\mathbf{x}}), \cdots, \mathbf{s}_n(\tilde{\mathbf{x}})$, *and* $\mathbf{s}_k(\tilde{\mathbf{x}})$ *represents the k-th order score of* $q_\sigma(\tilde{\mathbf{x}}) = \int p_{\text{data}}(\mathbf{x}) q_\sigma(\tilde{\mathbf{x}}|\mathbf{x}) d\mathbf{x}$.

Theorem 3.1 shows that there exists an equality between high-order moments of the posterior distribution of $\mathbf{x}$ given $\tilde{\mathbf{x}}$ and high-order scores with respect to $\tilde{\mathbf{x}}$. Leveraging Theorem 3.1 and the least squares estimation of $\mathbb{E}[\otimes^k \mathbf{x}|\tilde{\mathbf{x}}]$, the objectives for approximating the k-th order scores $\mathbf{s}_k(\tilde{\mathbf{x}})$ can be constructed as follows.

**Theorem 3.2.** *(Meng et al., 2021)* *Given score functions* $\mathbf{s}_1(\tilde{\mathbf{x}}), \ldots, \mathbf{s}_{k-1}(\tilde{\mathbf{x}})$, *a k-th order score model* $\mathbf{s}_k(\tilde{\mathbf{x}}; \boldsymbol{\theta})$ *can be obtained by optimizing the following objective:*

$$\boldsymbol{\theta}^* = \arg\min_{\boldsymbol{\theta}} \mathbb{E}_{p_{\text{data}}(\mathbf{x})} \mathbb{E}_{q_\sigma(\tilde{\mathbf{x}}|\mathbf{x})} \Big[$$
$$\| \otimes^k \mathbf{x} - f_k(\tilde{\mathbf{x}}, \mathbf{s}_1(\tilde{\mathbf{x}}), \ldots, \mathbf{s}_{k-1}(\tilde{\mathbf{x}}), \mathbf{s}_k(\tilde{\mathbf{x}}; \boldsymbol{\theta})) \|_2^2 \Big].$$

*where* $f_k$ *is a polynomial of* $\{\tilde{\mathbf{x}}, \mathbf{s}_1(\tilde{\mathbf{x}}), \ldots, \mathbf{s}_k(\tilde{\mathbf{x}})\}$ *such that:*

$$f_k(\tilde{\mathbf{x}}, \mathbf{s}_1(\tilde{\mathbf{x}}), \ldots, \mathbf{s}_k(\tilde{\mathbf{x}}))$$
$$= \begin{cases} \tilde{\mathbf{x}} + \sigma^2 \mathbf{s}_1(\tilde{\mathbf{x}}), & \text{if } k = 1, \\ \sigma^2 \frac{\partial}{\partial \tilde{\mathbf{x}}} f_{k-1} + \sigma^2 f_{k-1} \otimes \left( \mathbf{s}_1(\tilde{\mathbf{x}}) + \frac{\tilde{\mathbf{x}}}{\sigma^2} \right), & \text{if } k \geq 2. \end{cases} \quad (3)$$

*We have* $\mathbf{s}_k(\tilde{\mathbf{x}}; \boldsymbol{\theta}^*) = \mathbf{s}_k(\tilde{\mathbf{x}})$ *for almost all* $\tilde{\mathbf{x}}$.

## 3.2. High-Order DSM with Missing Data

Let $\mathbf{x} = (x_1, x_2, \ldots, x_d) \in \mathbb{R}^d$ be a random vector sampled from an unknown data distribution $p_{\text{data}}(\mathbf{x})$, and $\mathbf{m} = (m_1, m_2, \ldots, m_d) \in \{0, 1\}^d$ be a binary mask where $m_i = 1$ indicates that $x_i$ is missing, and $m_i = 0$ indicates that $x_i$ is observed. The observed data can be express as $\mathbf{x}_{\text{obs}} = \mathbf{x} \odot (1 - \mathbf{m}) + \text{na} \odot \mathbf{m}$, where $\odot$ denotes element-wise multiplication, and na indicates the missing value. Perturbing $\mathbf{x}$ with Gaussian noise results in $\tilde{\mathbf{x}}|\mathbf{x} \sim \mathcal{N}(\mathbf{x}, \sigma^2 \mathbf{I}_d)$, and the corresponding conditional density function is $q_\sigma(\tilde{\mathbf{x}}|\mathbf{x}) := (2\pi\sigma^2)^{-\frac{d}{2}} \exp\{-\frac{(\tilde{\mathbf{x}}-\mathbf{x})^\top(\tilde{\mathbf{x}}-\mathbf{x})}{2\sigma^2}\}$, where $\sigma$ is a pre-specified constant.

Rubin (1976) categorized missing data mechanisms into three types based on the dependency between the missing indicator ($\mathbf{m}$) and the complete data ($\mathbf{x}$): (1) Missing Completely at Random (MCAR), where $\mathbf{m}$ is independent of the complete data $\mathbf{x}$; (2) Missing at Random (MAR), where $\mathbf{m}$ depends only on the observed values $\mathbf{x}_{\text{obs}}$; and (3) Missing Not at Random (MNAR), where $\mathbf{m}$ depends on both the observed values $\mathbf{x}_{\text{obs}}$ and the missing values. Many previous efforts have concentrated on addressing the complexities of MNAR cases; however, M(C)AR cases have received less attention, as they allow for recovering the true distribution without the need for additional assumptions (Wang et al., 2019; Yang et al., 2019; Little & Rubin, 2019).

The MCAR mechanism simplifies modeling by eliminating bias introduced by missingness but is rarely realistic. MAR, on the other hand, is more applicable in real-world settings, as it allows missingness to depend on observed data. This dependency, however, can introduce bias in estimates if the missing data mechanism is not accounted for, making the problem considerably more challenging than in the MCAR framework. Inverse Probability Weighting

(IPW) has proven useful in correcting bias by assigning weights to observed data points inversely proportional to their probability of being observed (Wooldridge, 2007; Seaman & White, 2013). This ensures that observations more likely to be missing receive higher weights, enabling unbiased estimates that reflect the full data distribution and mitigating the bias introduced by the MAR mechanism.

The following theorem presents our first theoretical result, demonstrating that DSM with a missing indicator can recover the oracle score under M(C)AR mechanisms, defined as the gradient of $\log p_{\text{data}}(\mathbf{x})$ with respect to $\mathbf{x}$.

**Theorem 3.3.** *Let the missing mechanism of $\mathbf{x}$ be either MCAR or MAR, with the missing probability of every element lying between 0 and 1, i.e., $p(m_i = 1) \in [0,1)$ for all $i \in \{1, 2, \ldots, d\}$. Define the objective function:*

$$\mathcal{J}_{DSM}(\boldsymbol{\theta}) = \mathbb{E}_{\mathbf{x},\mathbf{m}}\mathbb{E}_{\tilde{\mathbf{x}}|\mathbf{x},\mathbf{m}}\left[\left\|\left\{\mathbf{s}_1(\tilde{\mathbf{x}};\boldsymbol{\theta})\right.\right.\right.$$
$$\left.\left.\left. + \frac{1}{\sigma^2}(\tilde{\mathbf{x}} - \mathbf{x})\right\} \odot \mathbf{w}_1\right\|_2^2\right],$$

*where the weight $\mathbf{w}_1$ is defined as:*

$$\mathbf{w}_1 = \begin{cases} 1 - \mathbf{m}, & \text{if MCAR}, \\ \frac{1-\mathbf{m}}{\sqrt{\mathbb{P}[\mathbf{m}=0|\mathbf{x}=\mathbf{x}]}}, & \text{if MAR}. \end{cases}$$

*If there exists a unique $\boldsymbol{\theta}^*$ such that $\mathbf{s}_1(\tilde{\mathbf{x}}) = \mathbf{s}_1(\tilde{\mathbf{x}};\boldsymbol{\theta}^*)$, then $\boldsymbol{\theta}^* = \arg\min_{\boldsymbol{\theta}} \mathcal{J}_{DSM}(\boldsymbol{\theta})$.*

It follows that the global optimum of DSM with a missing mask aligns with the oracle score under M(C)AR missing mechanisms. The detailed proof is provided in Appendix A.1. Extending this to the second-order score model, and building on Theorem 3.2, the second-order DSM with a missing mask can successfully recover the oracle second-order score, which corresponds to the Hessian of $\log p_{\text{data}}(\mathbf{x})$ with respect to $\mathbf{x}$.

**Theorem 3.4.** *Suppose the first-order score $\mathbf{s}_1(\tilde{\mathbf{x}})$ is given, and the missing mechanism of $\mathbf{x}$ is either MCAR or MAR, with $p(m_i = 1) \in [0,1)$ for all $i \in \{1, 2, \ldots, d\}$. Define the objective function:*

$$\mathcal{J}_{D_2SM}(\boldsymbol{\theta}) = \mathbb{E}_{\mathbf{x},\mathbf{m}}\mathbb{E}_{\tilde{\mathbf{x}}|\mathbf{x},\mathbf{m}}\left[\left\|\left\{\mathbf{s}_2(\tilde{\mathbf{x}};\boldsymbol{\theta}) + \mathbf{s}_1(\tilde{\mathbf{x}})\mathbf{s}_1^\top(\tilde{\mathbf{x}})\right.\right.\right.$$
$$\left.\left.\left. + \frac{\mathbf{I} - \mathbf{z}\mathbf{z}^\top}{\sigma^2}\right\} \odot \mathbf{w}_2\right\|_2^2\right],$$

*where $\mathbf{z} = \frac{\tilde{\mathbf{x}}-\mathbf{x}}{\sigma}$, and $\mathbf{w}_2$ is defined as:*

$$\mathbf{w}_2 = \begin{cases} (1-\mathbf{m})(1-\mathbf{m})^\top, & \text{if MCAR}, \\ \frac{(1-\mathbf{m})(1-\mathbf{m})^\top}{\sqrt{\mathbb{P}[\mathbf{m}\mathbf{m}^\top=0|\mathbf{x}]}}, & \text{if MAR}. \end{cases}$$

*If there exists a unique $\boldsymbol{\theta}^*$ exists such that $\mathbf{s}_2(\tilde{\mathbf{x}}) = \mathbf{s}_2(\tilde{\mathbf{x}};\boldsymbol{\theta}^*)$, then $\boldsymbol{\theta}^* = \arg\min_{\boldsymbol{\theta}} \mathcal{J}_{D_2SM}(\boldsymbol{\theta})$.*

We can conclude that the global optimum of the second-order DSM with a missing mask corresponds to the oracle second-order score. A detailed proof is provided in Appendix A.2. We now extend to any desired order under M(C)AR mechanisms. Theorem 3.5 indicates that $k$-th order DSM with missing mask can learn the oracle $k$-th order score and the proof is provided in Appendix A.3.

**Theorem 3.5.** *Given score functions $\mathbf{s}_1(\tilde{\mathbf{s}}), \cdots, \mathbf{s}_{k-1}(\tilde{\mathbf{s}})$, and the missing probability of every element lies between 0 and 1, which is $p(m_i = 1) \in [0,1)$ for all $i \in \{1, 2, \ldots, d\}$. If we correctly model the $k-$th order derivative $\mathbf{s}_k(\tilde{\mathbf{x}})$, there exists $\boldsymbol{\theta}^*$ such that $\mathbf{s}_k(\tilde{\mathbf{x}}, \boldsymbol{\theta}^*) = \mathbf{s}_k(\tilde{\mathbf{x}})$, then*

$$\boldsymbol{\theta}^* = \arg\min_{\boldsymbol{\theta}} \mathbb{E}_{\tilde{\mathbf{x}},\mathbf{x},\mathbf{m}}\left[\left\|\left\{\otimes^k \mathbf{x} - f_k(\tilde{\mathbf{x}}, \mathbf{s}_1(\tilde{\mathbf{x}}), \ldots, \mathbf{s}_{k-1}(\tilde{\mathbf{x}}),\right.\right.\right.$$
$$\left.\left.\left. \mathbf{s}_k(\tilde{\mathbf{x}};\boldsymbol{\theta}))\right\} \odot \otimes^k \mathbf{w}\right\|^2\right],$$

*where $\mathbf{w} = \mathbf{1} - \mathbf{m}$ if the missing mechanism of $\mathbf{x}$ is MCAR, and $\mathbf{w} = \frac{\mathbf{1}-\mathbf{m}}{\mathbb{P}[\mathbf{m}^k=0|\mathbf{x}=\mathbf{x}]}$ if the missing mechanism of $\mathbf{x}$ is MAR.*

## 4. Training Score Models by High-Order DSM with Missing data

In this section, we describe the training process for high-order score models in the presence of missing data and evaluate their empirical performance. While our analysis specifically focuses on the first- and second-order scores, the approach can be applied to any order of scores.

As derived in Theorem 3.3, the first-order score model $\mathbf{s}_1(\tilde{\mathbf{x}};\boldsymbol{\theta})$ is learned by minimizing the following objective function using the observed data,

$$\mathcal{L}_{\text{DSM}}(\boldsymbol{\theta}) = \mathbb{E}_{\tilde{\mathbf{x}}_{\text{obs}},\mathbf{x}_{\text{obs}},\mathbf{m}}\left[\left\|\left\{\mathbf{s}_1(\tilde{\mathbf{x}}_{\text{obs}};\boldsymbol{\theta})\right.\right.\right.$$
$$\left.\left.\left. + \frac{1}{\sigma^2}(\tilde{\mathbf{x}}_{\text{obs}} - \mathbf{x}_{\text{obs}})\right\} \odot \mathbf{w}_1\right\|_2^2\right]. \tag{4}$$

Similarly, the second-order score model $\mathbf{s}_2(\tilde{\mathbf{x}};\boldsymbol{\theta})$ is learned using the following objective, as derived in Theorem 3.4,

$$\mathcal{L}_{\text{D}_2\text{SM}}(\boldsymbol{\theta}) = \mathbb{E}_{\tilde{\mathbf{x}}_{\text{obs}},\mathbf{x}_{\text{obs}},\mathbf{m}}\left[\left\|\left\{\mathbf{s}_2(\tilde{\mathbf{x}}_{\text{obs}};\boldsymbol{\theta}) + \mathbf{s}_1(\tilde{\mathbf{x}}_{\text{obs}})\mathbf{s}_1^\top(\tilde{\mathbf{x}}_{\text{obs}})\right.\right.\right.$$
$$\left.\left.\left. + \frac{\mathbf{I} - \mathbf{z}_{\text{obs}}\mathbf{z}_{\text{obs}}^\top}{\sigma^2}\right\} \odot \mathbf{w}_2\right\|_2^2\right]. \tag{5}$$

However, training the second-order score model, $\mathbf{s}_2(\tilde{\mathbf{x}};\boldsymbol{\theta})$, requires knowledge of the first-order score $\mathbf{s}_1(\tilde{\mathbf{x}})$. Therefore,

we adopt a multi-task objective to train both $\mathbf{s}_1(\tilde{\mathbf{x}}; \boldsymbol{\theta})$ and $\mathbf{s}_2(\tilde{\mathbf{x}}; \boldsymbol{\theta})$ simultaneously,

$$\mathcal{L}_{\text{joint}}(\boldsymbol{\theta}) = \mathcal{L}_{\text{DSM}}(\boldsymbol{\theta}) + \omega \mathcal{L}_{\text{D}_2\text{SM}}(\boldsymbol{\theta}), \tag{6}$$

where $\omega \in \mathbb{R}^+$ is a tunable coefficient. $\mathbb{P}[\mathbf{m} = 0 | \mathbf{x} = \mathbf{x}_{\text{obs}}]$ in MCAR and $\mathbb{P}[\mathbf{mm}^\top = 0 | \mathbf{x} = \mathbf{x}_{\text{obs}}]$ in MAR are estimated using logistic regression models, where the response variable indicates whether the data is missing or observed, and the predictors are the observed variables. In the experiments, missing values are handled by replacing them with $0$ for continuous variables and creating a new category for discrete variables. One-hot encoding is then applied to discrete variables. Element-wise multiplication with the mask naturally mitigates the impact of replacing missing values with zeros when computing the objective. The algorithm is provided in Appendix B.

## 4.1. Improving stability with Variance Reduction

It is important to note that, in order to match the score of the true distribution $p_{\text{data}}(\mathbf{x})$, $\sigma$ needs to be close to zero for both DSM and D$_2$SM, so that $q_\sigma(\tilde{\mathbf{x}})$ closely approximates $p_{\text{data}}(\mathbf{x})$. However, training score models using denoising methods can suffer from high variance when $\sigma$ approaches zero. This challenge motivates the use of variance reduction techniques. Building on existing variance reduction methods for DSM (Song & Kingma, 2021; Meng et al., 2021), we propose tailored variance reduction techniques specifically for training DSM with missing data, as follows,

$$\mathcal{L}_{\text{DSM-VR}}(\boldsymbol{\theta}) = \mathcal{L}_{\text{DSM}}(\boldsymbol{\theta}) - \mathbb{E}_{\mathbf{x}_{\text{obs}}, \mathbf{m}} \mathbb{E}_{\mathbf{z} \sim \mathcal{N}(\mathbf{0}, \mathbf{I})} \left[ \left( \frac{2}{\sigma} \mathbf{s}(\mathbf{x}_{\text{obs}}; \boldsymbol{\theta})^\top \mathbf{z} \right) \right.$$
$$\left. \odot \mathbf{g}_1(\mathbf{x}_{\text{obs}}, \mathbf{m}) + \frac{\|\mathbf{z} \odot \mathbf{g}_1(\mathbf{x}_{\text{obs}}, \mathbf{m})\|^2}{\sigma^2} \right], \tag{7}$$

where $\mathbf{g}_1(\mathbf{x}_{\text{obs}}, \mathbf{m}) = \mathbf{1} - \mathbf{m}$ under MCAR, and $\mathbf{g}_1(\mathbf{x}_{\text{obs}}, \mathbf{m}) = \frac{\mathbf{1} - \mathbf{m}}{\sqrt{\mathbb{P}[\mathbf{m} = 0 | \mathbf{x} = \mathbf{x}_{\text{obs}}]}}$ under MAR.

For the second-order model with missing data, we implement a variance reduction (VR) technique using antithetic sampling (James, 1985; Meng et al., 2021), which involves utilizing two negatively correlated sample vectors centered around $\mathbf{x}$. The objective function is then formulated as

$$\mathcal{L}_{\text{D}_2\text{SM-VR}}(\boldsymbol{\theta}) = \mathbb{E}_{\mathbf{x}_{\text{obs}}, \mathbf{m}} \mathbb{E}_{\mathbf{z} \sim \mathcal{N}(\mathbf{0}, \mathbf{I})} \left[ \left\{ \psi(\tilde{\mathbf{x}}_{\text{obs}}^+)^2 + \psi(\tilde{\mathbf{x}}_{\text{obs}}^-)^2 \right. \right.$$
$$\left. \left. + 2\frac{\mathbf{I} - \mathbf{z}\mathbf{z}^\top}{\sigma} \odot \boldsymbol{\Psi} \right\} \odot \mathbf{g}_2(\mathbf{x}_{\text{obs}}, \mathbf{m}) \right], \tag{8}$$

where the antithetic samples are defined as $\mathbf{x}_{\text{obs}}^+ = \mathbf{x}_{\text{obs}} + \sigma\mathbf{z}$ and $\mathbf{x}_{\text{obs}}^- = \mathbf{x}_{\text{obs}} - \sigma\mathbf{z}$. Here, $\boldsymbol{\psi} = \mathbf{s}_2 + \mathbf{s}_1\mathbf{s}_1^\top$, and $\boldsymbol{\Psi} = \left( \psi(\tilde{\mathbf{x}}_{\text{obs}}^+) + \psi(\tilde{\mathbf{x}}_{\text{obs}}^-) - 2\psi(\mathbf{x}_{\text{obs}}) \right)$. Under the MCAR

setting, $\mathbf{g}_2(\mathbf{x}_{\text{obs}}, \mathbf{m}) = (\mathbf{1} - \mathbf{m})(\mathbf{1} - \mathbf{m})^\top$, while for MAR, $\mathbf{g}_2(\mathbf{x}_{\text{obs}}, \mathbf{m}) = \frac{(\mathbf{1} - \mathbf{m})(\mathbf{1} - \mathbf{m})^\top}{\sqrt{\mathbb{P}[\mathbf{mm}^\top = 0 | \mathbf{x} = \mathbf{x}]}}$. The formal analysis of variance reduction is provided in sections A.4 and A.5 of the Appendix.

We perform an empirical analysis to assess the impact of VR on training score models with DSM and D$_2$SM using incomplete data. The full data is generated from a 2-d Gaussian distribution and we simulate the incomplete data under MCAR, training $\mathbf{s}_1(\tilde{\mathbf{x}}_{\text{obs}}; \boldsymbol{\theta})$ and $\mathbf{s}_2(\tilde{\mathbf{x}}_{\text{obs}}; \boldsymbol{\theta})$ using a joint learning objective Eq. (6). Using a sample size of 1000 and a missing ratio of 0.3, In Figure 1, we compare the estimated score and Hessian for the first dimension against the ground truth at noise levels $\sigma = \{0.1, 0.001\}$, both with and without VR. The results indicate that VR is essential for accurate estimation in both DSM and D$_2$SM when $\sigma$ is close to zero, while its importance diminishes at higher values of $\sigma$. As $\sigma$ increases, both methods still achieve reasonable score estimates even without VR. Additionally, when using complete data and varying the missing ratio $\alpha = \{0.1, 0.3, 0.5\}$ with DSM and VR at $\sigma = 0.001$, we observe that the first- and second-order score estimates remain close to the ground truth. While performance degrades as the proportion of missing data increases, the estimates remain generally accurate.

## 4.2. Scalability and Numerical Stability

We show that the proposed method efficiently and accurately estimates second-order scores across different missing ratios, as summarized in Table 1. To achieve this, we generate 10 synthetic datasets with known ground truth, consisting of 100-dimensional correlated multivariate normal distributions that include varying levels of missing data under MCAR mechanism. The covariance matrix for these distributions is constructed using eigenvalues $t \in \{1, 5\}$ to vary degrees of correlation. We evaluate the performance of the estimated $\mathbf{s}_1$ and the diagonal of $\mathbf{s}_2$ by calculating the mean squared error (MSE) between the estimated scores and the ground truth scores derived from the complete data, across various values of $\sigma$. Our results indicate that the jointly optimized $\mathbf{s}_1(\tilde{\mathbf{x}}_{\text{obs}}; \boldsymbol{\theta})$ and $\mathbf{s}_2(\tilde{\mathbf{x}}_{\text{obs}}; \boldsymbol{\theta})$ achieve empirical performance close to the ground truth. As previously mentioned, when $\sigma$ approaches zero, the estimates without VR become unreliable for both the score and Hessian, likely due to convergence issues. Although performance declines with an increase in missing data, the estimates remain reasonable.

## 5. Sampling with Missing Data via Second-Order Score Models

In this section, we illustrate how our second-order score model $\mathbf{s}_2(\tilde{\mathbf{x}}; \boldsymbol{\theta})$, trained on incomplete data, improves sam-

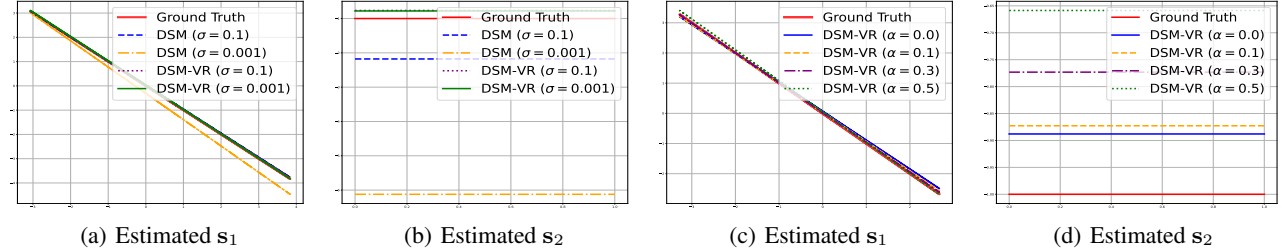

| (a) Estimated $\mathbf{s}_1$ | (b) Estimated $\mathbf{s}_2$ | (c) Estimated $\mathbf{s}_1$ | (d) Estimated $\mathbf{s}_2$ |

*Figure 1.* Comparison of estimated $\mathbf{s}_1$ and $\mathbf{s}_2$ under different conditions. (a) and (b) show estimates with DSM and D$_2$SM varying the noise level $\sigma$ with a fixed missing ratio 0.3. (c) and (d) show estimates with DSM and D$_2$SM varying the missing ratio $\alpha$.

*Table 1.* Mean squared error (MSE) between the estimated first-order and second-order scores and the ground truth is evaluated across 5,000 test samples. We vary the noise scales $\sigma$ and missing ratios $\alpha$, with each configuration tested using 10 random seeds.

| Methods | $\alpha = 0.0$ | | $\alpha = 0.1$ | | $\alpha = 0.3$ | | $\alpha = 0.5$ | |
|---|---|---|---|---|---|---|---|---|
| | $\sigma = 0.1$ | $\sigma = 0.01$ | $\sigma = 0.1$ | $\sigma = 0.01$ | $\sigma = 0.1$ | $\sigma = 0.01$ | $\sigma = 0.1$ | $\sigma = 0.01$ |
| $\mathbf{s}_1$ | $0.28 \pm 0.01$ | $0.42 \pm 0.02$ | $0.29 \pm 0.01$ | $0.42 \pm 0.02$ | $0.32 \pm 0.01$ | $0.44 \pm 0.01$ | $0.37 \pm 0.01$ | $0.44 \pm 0.02$ |
| $\mathbf{s}_1$(VR) | $0.07 \pm 0.00$ | $0.07 \pm 0.00$ | $0.09 \pm 0.00$ | $0.09 \pm 0.00$ | $0.13 \pm 0.00$ | $0.15 \pm 0.01$ | $0.23 \pm 0.00$ | $0.27 \pm 0.01$ |
| $\mathbf{s}_2$ | $0.16 \pm 0.02$ | $15.42 \pm 0.47$ | $0.16 \pm 0.02$ | $27.42 \pm 2.34$ | $0.16 \pm 0.02$ | $29.74 \pm 3.35$ | $0.17 \pm 0.03$ | $26.08 \pm 2.03$ |
| $\mathbf{s}_2$(VR) | $0.04 \pm 0.00$ | $0.05 \pm 0.00$ | $0.04 \pm 0.00$ | $0.04 \pm 0.00$ | $0.04 \pm 0.00$ | $0.05 \pm 0.00$ | $0.05 \pm 0.00$ | $0.06 \pm 0.01$ |

ple quality and enables faster convergence in terms of the number of sampling steps required. We demonstrate the effectiveness of the proposed model through simulations and real-world datasets containing missing values, comparing its performance against various baseline methods.

**Langevin dynamics.** Langevin dynamics samples data from $p_{\text{data}}(\mathbf{x})$ by utilizing the first-order score function $\mathbf{s}_1(\mathbf{x})$ (Bussi & Parrinello, 2007; Song & Ermon, 2019). Starting with a prior distribution $\pi(\mathbf{x})$, a fixed step size $\epsilon > 0$, and an initial value $\tilde{\mathbf{x}}_0 \sim \pi(\mathbf{x})$, Langevin dynamics iteratively updates the samples as follows:

$$\tilde{\mathbf{x}}_{t+1} = \tilde{\mathbf{x}}_t + \frac{1}{2}\epsilon\mathbf{s}_1(\tilde{\mathbf{x}}_t) + \sqrt{\epsilon}\mathbf{z}_t, \qquad (9)$$

where $\mathbf{z}_t \sim \mathcal{N}(\mathbf{0}, \mathbf{I})$ represents Gaussian noise.

**Ozaki Sampling.** Following Meng et al. (2021), Ozaki discretization improves data synthesis by integrating second-order information from $\mathbf{s}_2(\mathbf{x})$ to precondition the sampling process. The updates for Ozaki sampling are performed as follows:

$$\tilde{\mathbf{x}}_t = \tilde{\mathbf{x}}_{t-1} + \mathbf{M}_{t-1}\mathbf{s}_1(\tilde{\mathbf{x}}_{t-1}) + \Sigma_{t-1}^{1/2}\mathbf{z}_t, \qquad (10)$$

where $\mathbf{z}_t \sim \mathcal{N}(\mathbf{0}, \mathbf{I})$, $\mathbf{M}_{t-1} = e^{\epsilon\mathbf{s}_2(\tilde{\mathbf{x}}_{t-1})} - \mathbf{I}$, and $\Sigma_{t-1} = (e^{2\epsilon\tilde{\mathbf{x}}_{t-1}} - \mathbf{I})\mathbf{s}_2(\tilde{\mathbf{x}}_{t-1})^{-1}$.

**Illustration.** We use the Swiss-Roll dataset to demonstrate the effectiveness of Ozaki Sampling, focusing on its convergence rate and quality of data generation through second-order information. Both methods employ a step size of $\epsilon = 0.005$ and a missing ratio of 0.5 under the MCAR missing mechanism. As shown in Figure 2, Ozaki Sampling

generates comparable data to Langevin dynamics with fewer iterations, resulting in data that is more concentrated around the original distribution, while Langevin dynamics yields noisier and more dispersed results.

Following Kim et al. (2022); Ouyang et al. (2023), we conduct experiments on a simulated Bayesian Network dataset and a real Census dataset (Kohavi, 1996) to illustrate the efficiency and effectiveness of the data generated by our proposed model trained on missing data.

**Baselines.** We evaluate the proposed method using both Langevin and Ozaki sampling against several baseline techniques for synthetic data generation on datasets with missing values. Specifically, we implement (1) a vanilla DSM model that removes rows with missing values (termed DSM-delete), and (2) STaSy (Kim et al., 2022), a state-of-the-art score-based model, which significantly outperforms other approaches for tabular data (termed STaSy-mean). Since STaSy requires complete datasets for training, we apply mean imputation to handle any missing values in the training data.

**Metrics.** Following Kim et al. (2022); Ouyang et al. (2023), we employ two criteria, *fidelity* and *machine learning (ML) efficiency*, to assess the quality of the generated synthetic tabular data. For evaluating *fidelity*, we utilize the model-agnostic library SDMetrics. The result ranges from 0 to 100%. A higher score indicates better overall quality of the synthetic data. To measure *ML efficiency*, we adopt the same pipeline as Kim et al. (2022); Kotelnikov et al. (2023), training various models—including Decision Tree, AdaBoost, Logistic Regression, MLP Classifier, Random

Forest, and XGBoost on the synthetic data and testing them with real data. Our primary metric is classification accuracy, and we also report AUROC and Weighted-F1 scores in the Appendix C.4. All experimental results are based on three repetitions.

**Results.** Figure 5 and Table 2 demonstrate the effectiveness of the proposed method on both the simulated Bayesian Network dataset and the Census data, showing superior performance in terms of fidelity and utility compared to other baselines. Specifically, these results confirm that the impute-then-generate approach introduces bias, whereas directly learning from missing data significantly improves the performance of the generative model. Furthermore, the advantages of the proposed model become more pronounced as the missing ratios increase. Additional details and results of the experiments can be found in Appendix C.

*Table 2. Fidelity and ML efficiency evaluation of MissScore using Langevin and Ozaki samplings, along with other baselines, on the Census dataset with a missing ratio of 0.3 under MCAR.*

|  | Langevin | Ozaki | DSM-delete | STaSy-mean |
|---|---|---|---|---|
| *Fidelity* | 86% | **88**% | 73% | 82% |
| *ML efficiency* | 80% | **81**% | 70% | 77% |

# 6. Causal Discovery with Missing Data via Second-Order Scores

**Background.** Causal discovery aims to identify causal relationship from purely observational data. However, the task is ill-posed without additional assumptions. Assuming an additive noise model (ANM) allows for the identification of causal structures. In this context, consider the ANM defined as $x_i = f_i(x_{\mathrm{PA}_i}) + z_i$, where $f_i$ is a nonlinear function and $z_i$ is a Gaussian noise. Rolland et al. (2022) proposed an order-based algorithm that uses the second order score of an ANM with a probability distribution $p_{\mathrm{data}}(\mathbf{x})$ to identify leaf nodes, and iteratively determine the topological order of the variables. However, the computation of the Hessian requires complete data, which poses challenges in real-world scenarios such as clinical trials, and biology, where missing data is common.

A straightforward approach to address missing data problem is to first impute the incomplete entries using off-the-shelf imputation methods and then apply existing causal discovery methods. However, this two-step approach can be suboptimal, as the imputation process may introduce bias for modeling the underlying data distribution. Our method mitigates this issue by directly training a second-order model with incomplete data, thereby reducing potential bias. Since the Hessian only provides information about variable order, we adopt a strategy similar to Rolland et al. (2022); Sanchez et al. (2022), first computing the topological order and then

using CAM pruning to derive the final directed acyclic graph (DAG) (Bühlmann et al., 2014).

**Baselines.** We utilize the MissForest imputation method to address missing data, followed by the implementation of DiffAN (Sanchez et al., 2022) (termed MissDiffAN) and DAGMA (Bello et al., 2022) (termed MissForest) for structure learning. DiffAN serves as a diffusion-based adaptation of the approach proposed by (Rolland et al., 2022), ensuring a fair comparison. Furthermore, we compare MissScore with MissDAG (Gao et al., 2022) and MissOTM (Vo et al., 2024), both of which are prominent methods that have shown superior performance in causal discovery with missing data relative to various other baselines.

**Metrics.** All quantitative results are averaged over 10 random initiazations. For comparing the estimated DAG with the ground-truth one, we report commonly used metrics: Order Divergence and Structural Hamming Distance (SHD). Order Divergence measures the number of errors in the ordering, while SHD indicates the minimum number of edge additions, deletions, and reversals needed to convert the recovered DAG into the true one. Lower values for both metrics are preferred. Order Divergence is calculated only for MissScore and MissDiffAN, as these are the only order-based methods.

**Simulations.** We simulate synthetic datasets generating a ground-truth DAG from the graph model Erdős–Rényi (ER). Each function $f_i$ is constructed from a multi-layer perceptron (MLP) and a multiple index model (MIM) with random coefficients. We consider a general scenario of non-euqal variances, sampling 1000 observations according to all missing mechansims: MCAR, MAR, MNAR at 10% and 30% missing rates and complete data. In the main text, we report the SHD and runtime in MCAR cases with missing ratio 0.1 varies with number of dimensions, using Gaussian noise. Specifically, the number of edges is set equal to the dimensionality.

**Results.** In Figure 4, our approach demonstrates comparable performance to the state-of-the-art methods MissDAG and MissOTM, although it shows slightly lower performance in the high-dimensional scenario with $d = 50$. This may be attributed to the challenges of training a denoising score matching model with a limited number of samples. However, MissScore significantly reduces computation time compared to MissOTM and MissDAG, offering an efficient alternative while maintaining similar results. With MissForest, our performance is similar, possibly due to the additional constraints enforced during training by DAGMA. However, when comparing our method to those that rely on imputation followed by causal discovery approach, we find that our approach outperforms MissDiffAN. This suggests that imputation can introduce bias in downstream tasks. Additional results for various missing ratios, mechanisms, and order

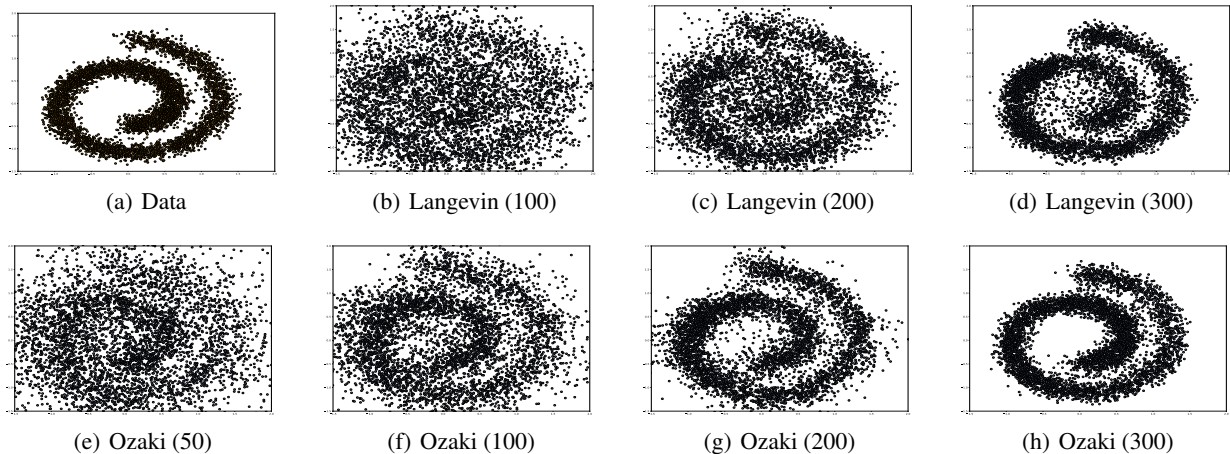

(a) Data     (b) Langevin (100)     (c) Langevin (200)     (d) Langevin (300)

(e) Ozaki (50)     (f) Ozaki (100)     (g) Ozaki (200)     (h) Ozaki (300)

*Figure 2.* Sampling a Swiss Roll dataset with a step size of $0.005$ using Langevin dynamics and Ozaki sampling. Ozaki sampling achieves higher-quality samples with fewer iterations compared to Langevin dynamics under MCAR with a missing ratio of $0.5$. Figure (a) displays the dataset; Figures (b)-(d) show the results from Langevin dynamics; and Figures (e)-(h) present the results from Ozaki sampling. The numbers in parentheses indicate the number of iterations taken.

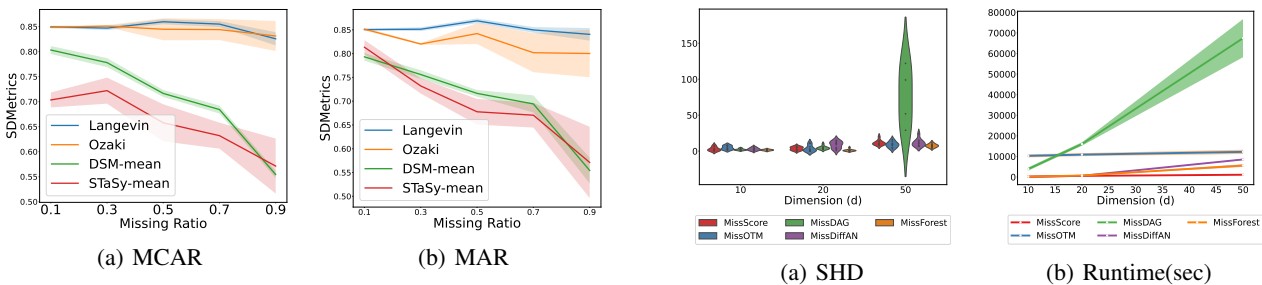

(a) MCAR     (b) MAR         (a) SHD     (b) Runtime(sec)

*Figure 3. Fidelity* evaluation of MissScore using Langevin and Ozaki samplings, along with other baselines, on the Bayesian Network dataset with varying missing ratios $\alpha = \{0.1, 0.3, 0.5, 0.7, 0.9\}$ under different missing mechanisms.

*Figure 4.* Data is generated under MCAR with a missing ratio of $0.1$, varying dimensions $d = \{10, 20, 50\}$ using ER graph model. The sample size is 1000, and $f_i$ corresponds to an MLP. Left: SHD; Right: Runtime. The shaded area indicates 95% confidence.

divergences are provided, along with further experimental details, in Appendix D.4.

## 7. Conclusion and Future Work

In this work, we present a novel framework for directly estimating high-order data density scores in the presence of missing data using DSM under M(C)AR mechanisms. Unlike methods that rely on imputation or additional model training, our approach handles missing data directly and efficiently. Distinct from prior work like MissDiff, MissScore focuses on high-order score learning and introduces an oracle estimator under the MAR mechanism. Empirical results demonstrate the efficiency and accuracy of our framework in estimating second-order scores, with applications showcasing improved sampling quality through Ozaki discretization in sernarios with missing data. Furthermore, our causal discovery method scales with dimensionality and achieves competitive results against state-of-the-art approaches.

Despite these advancements, certain limitations remain. The effectiveness of high-order score estimation diminishes in low-noise environments without variance reduction, particularly in scenarios with high levels of missingness. These issues also extend to downstream tasks, such as causal discovery and sampling, where performance tends to decline slightly with increasing dimensionality. Additionally, direct testing on real-world data poses a challenge due to the unavailability of ground truth.

Future work could explore extending high-order score estimation to domains such as image and time-series data, while investigating the application of DSM to directly handle MNAR data. Additionally, incorporating constraints into score-based models to enhance causal discovery with missing data holds significant potential for advancing score-based learning in complex and missing data scenarios.

## Acknowledgements

This research was supported by the University of Melbourne's Research Computing Services and the Petascale Campus Initiative. WL was supported by a Melbourne Research Scholarship. EG was supported by the Centre for Augmented Reasoning (CAR). QK was supported by ARC grant DE250100030. MG was supported by ARC grants DE210101624 and DP240102088, as well as WIS-MBZUAI grant 142571.

## Impact Statement

This paper aims to advance the field of machine learning by presenting a novel framework for high-order score estimation in the presence of missing data. Our work has the potential to enhance applications in data synthesis, generative modeling and causal discovery, which are crucial in domains such as healthcare, finance, and social sciences. By directly addressing the challenges posed by missing data, our framework offers an alternative to traditional imputation-based methods, enabling more accurate and unbiased modeling of incomplete datasets. Our contribution aligns with ongoing efforts to improve the ability of machine learning systems to operate under realistic and challenging data conditions, promoting more reliable, efficient, and equitable solutions across diverse applications. Additionally, the focus on high-order score estimation introduces new opportunities for improving sampling quality, uncertainty quantification, and the recovery of underlying causal structures, making this work applicable to both foundational research and practical implementations.

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

# Appendix

## Table of Contents

# A. Proofs

## A.1. Proof of Theorem 3.3

*Proof.* Under the MCAR missing mechanism, we know that

$$
\mathbb{E}_{\mathbf{x},\mathbf{m}}\mathbb{E}_{\tilde{\mathbf{x}}|\mathbf{x},\mathbf{m}}\left[\left\|\left\{\mathbf{s}(\tilde{\mathbf{x}};\boldsymbol{\theta})+\frac{1}{\sigma^2}(\tilde{\mathbf{x}}-\mathbf{x})\right\}\odot(\mathbf{1}-\mathbf{m})\right\|_2^2\right]
$$

$$
=\mathbb{E}_{\tilde{\mathbf{x}},\mathbf{x},\mathbf{m}}\left[\left\|\left\{\mathbf{s}(\tilde{\mathbf{x}};\boldsymbol{\theta})+\frac{1}{\sigma^2}(\tilde{\mathbf{x}}-\mathbf{x})\right\}\odot(\mathbf{1}-\mathbf{m})\right\|_2^2\right]
$$

$$
=\mathbb{E}_{\tilde{\mathbf{x}},\mathbf{x}}\mathbb{E}_{\mathbf{m}|\tilde{\mathbf{x}},\mathbf{x}}\left[\left\|\left\{\mathbf{s}(\tilde{\mathbf{x}};\boldsymbol{\theta})+\frac{1}{\sigma^2}(\tilde{\mathbf{x}}-\mathbf{x})\right\}\odot(\mathbf{1}-\mathbf{m})\right\|_2^2\right]
$$

$$
=\mathbb{E}_{\tilde{\mathbf{x}},\mathbf{x}}\left[\left\|\left\{\mathbf{s}(\tilde{\mathbf{x}};\boldsymbol{\theta})+\frac{1}{\sigma^2}(\tilde{\mathbf{x}}-\mathbf{x})\right\}\odot\sqrt{\mathbb{E}_{\mathbf{m}|\tilde{\mathbf{x}},\mathbf{x}}(\mathbf{1}-\mathbf{m})}\right\|_2^2\right]
$$

$$
=\mathbb{E}_{\tilde{\mathbf{x}},\mathbf{x}}\left[\left\|\left\{\mathbf{s}(\tilde{\mathbf{x}};\boldsymbol{\theta})+\frac{1}{\sigma^2}(\tilde{\mathbf{x}}-\mathbf{x})\right\}\odot\sqrt{\mathbf{w_1}}\right\|_2^2\right].
$$

Denote $q_\sigma(\tilde{\mathbf{x}}|\mathbf{x})=\mathcal{N}(\tilde{\mathbf{x}};\mathbf{x},\sigma^2)$, we only need to show there exists some constant $C$ independent of $\boldsymbol{\theta}$ such that

$$
\mathbb{E}_{\tilde{\mathbf{x}},\mathbf{x}}\left[\|\{\mathbf{s}(\tilde{\mathbf{x}};\boldsymbol{\theta})-\nabla_{\tilde{\mathbf{x}}}q_\sigma(\tilde{\mathbf{x}}|\mathbf{x})\}\odot\sqrt{\mathbf{w_1}}\|_2^2\right]=\mathbb{E}_{\tilde{\mathbf{x}},\mathbf{x}}\left[\|\{\mathbf{s}(\tilde{\mathbf{x}};\boldsymbol{\theta})-\mathbf{s}(\tilde{\mathbf{x}})\}\odot\sqrt{\mathbf{w_1}}\|_2^2\right]+C \tag{11}
$$

For simplicity, we consider the case $d=1$. For the right hand side of Eq.(11),

$$
\text{R.H.S}=w_1\cdot\left\{\mathbb{E}_{\tilde{x},x}\left[s^2(\tilde{x};\theta)\right]-2\mathbb{E}_{\tilde{x},x}\left[s(\tilde{x};\theta)s(\tilde{x})\right]\right\}+C.
$$

For the left hand side of Eq.(11),

$$
\text{L.H.S}=w_1\cdot\left\{\mathbb{E}_{\tilde{x},x}\left[s^2(\tilde{x};\theta)\right]-2\mathbb{E}_{\tilde{x},x}\left[s(\tilde{x};\theta)\nabla_{\tilde{x}}\log q_\sigma(\tilde{x}|x)\right]\right\}+C.
$$

Hence, we only need to show

$$
\mathbb{E}_{\tilde{x},x}\left[s(\tilde{x};\theta)s(\tilde{x})\right]=\mathbb{E}_{\tilde{x},x}\left[s(\tilde{x};\theta)\nabla_{\tilde{x}}\log q_\sigma(\tilde{x}|x)\right]. \tag{12}
$$

We have,

$$
\mathbb{E}_{\tilde{x},x}\left[s(\tilde{x};\theta)s(\tilde{x})\right]=\int s(\tilde{x};\theta)\nabla_{\tilde{x}}p_{\tilde{x}}(\tilde{x})d\tilde{x}
$$

$$
=\int s(\tilde{x};\theta)\nabla_{\tilde{x}}\left(\int p(x)q_\sigma(\tilde{x}|x)dx\right)d\tilde{x}
$$

$$
=\int\int s(\tilde{x};\theta)p(x)\nabla_{\tilde{x}}\left(q_\sigma(\tilde{x}|x)\right)dxd\tilde{x}
$$

$$
=\int\int s(\tilde{x};\theta)p(x)q_\sigma(\tilde{x}|x)\nabla_{\tilde{x}}\left(\log q_\sigma(\tilde{x}|x)\right)dxd\tilde{x}
$$

$$
=\int\int s(\tilde{x};\theta)p_{\tilde{x},x}(\tilde{x},x)\nabla_{\tilde{x}}\left(\log q_\sigma(\tilde{x}|x)\right)dxd\tilde{x}=\mathbb{E}_{\tilde{x},x}\left[s(\tilde{x};\theta)\nabla_{\tilde{x}}\log q_\sigma(\tilde{x}|x)\right]
$$

Hence, we get our desired result under the MCAR missing mechanism. By similar argument in the proof for the MCAR missing mechanism, under the MAR missing mechanism, we have:

$$\mathbb{E}_{\mathbf{x},\mathbf{m}}\mathbb{E}_{\tilde{\mathbf{x}}|\mathbf{x},\mathbf{m}}\left[\left\|\left\{\mathbf{s}(\tilde{\mathbf{x}};\boldsymbol{\theta})+\frac{1}{\sigma^2}(\tilde{\mathbf{x}}-\mathbf{x})\right\}\odot\frac{\mathbf{1}-\mathbf{m}}{\sqrt{\mathbb{P}[\mathbf{m}=\mathbf{0}|\mathbf{x}=\mathbf{x}]}}\right\|_2^2\right]$$

$$=\mathbb{E}_{\tilde{\mathbf{x}},\mathbf{x},\mathbf{m}}\left[\left\|\left\{\mathbf{s}(\tilde{\mathbf{x}};\boldsymbol{\theta})+\frac{1}{\sigma^2}(\tilde{\mathbf{x}}-\mathbf{x})\right\}\odot\frac{\mathbf{1}-\mathbf{m}}{\sqrt{\mathbb{P}[\mathbf{m}=\mathbf{0}|\mathbf{x}=\mathbf{x}]}}\right\|_2^2\right]$$

$$=\mathbb{E}_{\tilde{\mathbf{x}},\mathbf{x}}\mathbb{E}_{m|\tilde{\mathbf{x}},\mathbf{x}}\left[\left\|\left\{\mathbf{s}(\tilde{\mathbf{x}};\boldsymbol{\theta})+\frac{1}{\sigma^2}(\tilde{\mathbf{x}}-\mathbf{x})\right\}\odot\frac{\mathbf{1}-\mathbf{m}}{\sqrt{\mathbb{P}[\mathbf{m}=\mathbf{0}|\mathbf{x}=\mathbf{x}]}}\right\|_2^2\right]$$

$$=\mathbb{E}_{\tilde{\mathbf{x}},\mathbf{x}}\left[\left\|\left\{\mathbf{s}(\tilde{\mathbf{x}};\boldsymbol{\theta})+\frac{1}{\sigma^2}(\tilde{\mathbf{x}}-\mathbf{x})\right\}\odot\frac{\sqrt{\mathbb{E}_{m|\tilde{\mathbf{x}},\mathbf{x}}(\mathbf{1}-\mathbf{m})}}{\sqrt{\mathbb{P}[\mathbf{m}=\mathbf{0}|\mathbf{x}=\mathbf{x}]}}\right\|_2^2\right]$$

$$=\mathbb{E}_{\tilde{\mathbf{x}},\mathbf{x}}\left[\left\|\left\{\mathbf{s}(\tilde{\mathbf{x}};\boldsymbol{\theta})+\frac{1}{\sigma^2}(\tilde{\mathbf{x}}-\mathbf{x})\right\}\right\|_2^2\right].$$

where the last equality comes from $\mathbb{E}_{\mathbf{m}|\tilde{\mathbf{x}},\mathbf{x}}(\mathbf{1}-\mathbf{m})=\mathbb{E}_{\mathbf{m}|\mathbf{x}}(\mathbf{1}-\mathbf{m})=\mathbb{P}[\mathbf{m}=\mathbf{0}|\mathbf{x}=\mathbf{x}]$. Then following the steps in the proof for the MCAR missing mechanism, taking $\mathbf{w}$ as a vector of 1, we get our desired result. $\square$

### A.2. Proof of Theorem 3.4

*Proof.* Under the MCAR missing mechanism, we know that

$$\mathbb{E}_{\tilde{\mathbf{x}},\mathbf{x},\mathbf{m}}\left[\left\|\left\{\mathbf{s}_2(\tilde{\mathbf{x}};\boldsymbol{\theta})+\mathbf{s}_1(\tilde{\mathbf{x}})\mathbf{s}_1^\top(\tilde{\mathbf{x}})+\frac{\mathbf{I}-\mathbf{z}\mathbf{z}^\top}{\sigma^2}\right\}\odot\left\{(\mathbf{1}-\mathbf{m})(\mathbf{1}-\mathbf{m})^\top\right\}\right\|_2^2\right]$$

$$=\mathbb{E}_{\tilde{\mathbf{x}},\mathbf{x}}\left[\left\|\left\{\mathbf{s}_2(\tilde{\mathbf{x}};\boldsymbol{\theta})+\mathbf{s}_1(\tilde{\mathbf{x}})\mathbf{s}_1^\top(\tilde{\mathbf{x}})+\frac{\mathbf{I}-\mathbf{z}\mathbf{z}^\top}{\sigma^2}\right\}\odot\sqrt{\mathbb{E}_{\mathbf{m}|\tilde{\mathbf{x}},\mathbf{x}}(\mathbf{1}-\mathbf{m})(\mathbf{1}-\mathbf{m})^\top}\right\|_2^2\right]$$

$$=\mathbb{E}_{\tilde{\mathbf{x}},\mathbf{x}}\left[\left\|\left\{\mathbf{s}_2(\tilde{\mathbf{x}};\boldsymbol{\theta})+\mathbf{s}_1(\tilde{\mathbf{x}})\mathbf{s}_1^\top(\tilde{\mathbf{x}})+\frac{\mathbf{I}-\mathbf{z}\mathbf{z}^\top}{\sigma^2}\right\}\odot\sqrt{\mathbf{w}_2}\right\|_2^2\right]$$

It is sufficient to show

$$\mathbb{E}_{\tilde{\mathbf{x}},\mathbf{x}}\left[\left\|\left\{\mathbf{s}_2(\tilde{\mathbf{x}};\boldsymbol{\theta})+\mathbf{s}_1(\tilde{\mathbf{x}})\mathbf{s}_1^\top(\tilde{\mathbf{x}})+\frac{\mathbf{I}-\mathbf{z}\mathbf{z}^\top}{\sigma^2}\right\}\odot\sqrt{\mathbf{w}_2}\right\|_2^2\right]$$
$$=\mathbb{E}_{\mathbf{x}}\mathbb{E}_{\tilde{\mathbf{x}}|\mathbf{x}}\left[\|\{\mathbf{s}_2(\tilde{\mathbf{x}};\boldsymbol{\theta})-\mathbf{s}_2(\tilde{\mathbf{x}})\}\odot\sqrt{\mathbf{w}_2}\|_2^2\right]+C. \tag{13}$$

with some constant $C$ independent of $\boldsymbol{\theta}$.

For the right hand side of Eq.(13),

$$\mathbb{E}_{\mathbf{x}}\mathbb{E}_{\tilde{\mathbf{x}}|\mathbf{x}}\left[\|\{\mathbf{s}_2(\tilde{\mathbf{x}};\boldsymbol{\theta})-\mathbf{s}_2(\tilde{\mathbf{x}})\}\odot\sqrt{\mathbf{w}_2}\|_2^2\right]$$

$$=\sum_{i=1}^d\sum_{j=1}^d w_{2,ij}\cdot\left\{\mathbb{E}_{\tilde{x},x}\left[s_{2,ij}^2(\tilde{x};\theta)\right]-2\mathbb{E}_{\tilde{x},x}\left[s_{2,ij}(\tilde{x};\theta)s_{2,ij}(\tilde{x})\right]+\mathbb{E}_{\tilde{x},x}\left[s_{2,ij}^2(\tilde{x})\right]\right\}$$

$$=\sum_{i=1}^d\sum_{j=1}^d w_{2,ij}\cdot\left\{\mathbb{E}_{\tilde{x},x}\left[s_{2,ij}^2(\tilde{x};\theta)\right]-2\mathbb{E}_{\tilde{x},x}\left[s_{2,ij}(\tilde{x};\theta)s_{2,ij}(\tilde{x})\right]+\mathbb{E}_{\tilde{x},x}\left[s_{2,ij}^2(\tilde{x})\right]\right\}+C_1$$

where $C_1$ is some constant independent of $\boldsymbol{\theta}$.

For the left hand side of Eq.(13), note that $\frac{\mathbf{I}-\mathbf{z}\mathbf{z}^\top}{\sigma^2} = -\left\{\nabla_{\tilde{\mathbf{x}}}^2 q_\sigma(\tilde{\mathbf{x}}|\mathbf{x})\right\} - \nabla_{\tilde{x}} q_\sigma(\tilde{\mathbf{x}}|\mathbf{x}) \cdot \left\{\nabla_{\tilde{\mathbf{x}}} q_\sigma(\tilde{\mathbf{x}}|\mathbf{x})\right\}^\top$, then we know that

$$
\mathbb{E}_{\tilde{\mathbf{x}},\mathbf{x}}\left[\left\|\left\{\mathbf{s}_2(\tilde{\mathbf{x}};\boldsymbol{\theta}) + \mathbf{s}_1(\tilde{\mathbf{x}})\mathbf{s}_1^\top(\tilde{\mathbf{x}}) + \frac{\mathbf{I}-\mathbf{z}\mathbf{z}^\top}{\sigma^2}\right\} \odot \sqrt{\mathbf{w}_2}\right\|_2^2\right]
$$

$$
= \sum_{i=1}^d \sum_{j=1}^d w_{2,ij}\Bigg(\mathbb{E}_{\tilde{x},x}\left[s_{2,ij}^2(\tilde{x};\theta)\right] + 2\mathbb{E}\left[s_{2,ij}^2(\tilde{x};\theta)s_{1,i}(\tilde{x})s_{1,j}(\tilde{x})\right]
$$

$$
- 2\mathbb{E}\left[s_{2,ij}(\tilde{x};\theta)\nabla_{\tilde{x}_i}\nabla_{\tilde{x}_j}\log q_\sigma(\tilde{x}|x)\right]
$$

$$
- 2\mathbb{E}\left[s_{2,ij}(\tilde{x};\theta)\nabla_{\tilde{x}_i}q_\sigma(\tilde{x}|x)\nabla_{\tilde{x}_j}q_\sigma(\tilde{x}|x)\right]\Bigg) + C_2.
$$

Comparing left and right hand side of Eq.(13), it is sufficient to show

$$
\mathbb{E}_{\tilde{x},x}\left[s_{2,ij}(\tilde{x};\theta)s_{2,ij}(\tilde{x})\right] = \mathbb{E}\left[s_{2,ij}(\tilde{x};\theta)\nabla_{\tilde{x}_i}\nabla_{\tilde{x}_j}\log q_\sigma(\tilde{x}|x)\right]
$$
$$
+ \mathbb{E}\left[s_2(\tilde{x};\theta)\nabla_{\tilde{x}_i}q_\sigma(\tilde{x}|x)\nabla_{\tilde{x}_j}q_\sigma(\tilde{x}|x)\right] - \mathbb{E}\left[s_2(\tilde{x};\theta)s_1^2(\tilde{x})\right]. \tag{14}
$$

We have

$$
\mathbb{E}_{\tilde{x},x}\left[s_{2,ij}(\tilde{x};\theta)s_{2,ij}(\tilde{x})\right] = \int s_{2,ij}(\tilde{x};\theta)s_{2,ij}(\tilde{x})p_{\tilde{x}}(\tilde{x})d\tilde{x}
$$

$$
= \int s_{2,ij}(\tilde{x};\theta)\left(\nabla_{\tilde{x}_i}\left\{\frac{\nabla_{\tilde{x}_j}p(\tilde{x})}{p(\tilde{x})}\right\}\right)p_{\tilde{x}}(\tilde{x})d\tilde{x}dx
$$

$$
= \int s_{2,ij}(\tilde{x};\theta)\left\{\frac{\nabla_{\tilde{x}_i}\nabla_{\tilde{x}_j}p(\tilde{x})}{p(\tilde{x})} - \frac{\nabla_{\tilde{x}_i}p(\tilde{x}) \cdot \nabla_{\tilde{x}_j}p(\tilde{x})}{p^2(\tilde{x})}\right\}p_{\tilde{x}}(\tilde{x})d\tilde{x}
$$

$$
= \int s_2(\tilde{x};\theta)\nabla_{\tilde{x}_i}\nabla_{\tilde{x}_j}p(\tilde{x})d\tilde{x} \tag{15}
$$

$$
- \int s_2(\tilde{x};\theta)\frac{\nabla_{\tilde{x}_i}p(\tilde{x}) \cdot \nabla_{\tilde{x}_j}p(\tilde{x})}{p(\tilde{x})}d\tilde{x}. \tag{16}
$$

**For Eq.(15):**

$$
\int s_2(\tilde{x};\theta)\nabla_{\tilde{x}_i}\nabla_{\tilde{x}_j}p_{\tilde{x}}(\tilde{x})d\tilde{x} = \int s_2(\tilde{x};\theta)\nabla_{\tilde{x}_i}\nabla_{\tilde{x}_j}\left\{\int q_\sigma(\tilde{x}|x)p_x(x)dx\right\}d\tilde{x}
$$

$$
= \iint s_2(\tilde{x};\theta)\left\{\nabla_{\tilde{x}_i}\nabla_{\tilde{x}_j}q_\sigma(\tilde{x}|x)\right\} \cdot p_x(x)d\tilde{x}dx
$$

$$
= \iint s_2(\tilde{x};\theta)]\left\{\nabla_{\tilde{x}_i}\nabla_{\tilde{x}_j}\log q_\sigma(\tilde{x}|x)\right\} \cdot q_\sigma(\tilde{x}|x)p_x(x)d\tilde{x}dx
$$

$$
+ \iint s_2(\tilde{x};\theta)\frac{\nabla_{\tilde{x}_i}q_\sigma(\tilde{x}|x)\nabla_{\tilde{x}_j}q_\sigma(\tilde{x}|x)}{q_\sigma(\tilde{x}|x)}p_x(x)d\tilde{x}dx
$$

$$
= \iint s_2(\tilde{x};\theta)\left\{\nabla_{\tilde{x}_i}\nabla_{\tilde{x}_j}\log q_\sigma(\tilde{x}|x)\right\}p_{x,\tilde{x}}(x,\tilde{x})d\tilde{x}dx
$$

$$
+ \iint s_2(\tilde{x};\theta)\left\{\frac{\nabla_{\tilde{x}_i}q_\sigma(\tilde{x}|x)}{q_\sigma(\tilde{x}|x)}\right\}\left\{\frac{\nabla_{\tilde{x}_j}q_\sigma(\tilde{x}|x)}{q_\sigma(\tilde{x}|x)}\right\}p_{x,\tilde{x}}(x,\tilde{x})d\tilde{x}dx
$$

$$
= \mathbb{E}\left[s_2(\tilde{x};\theta)\nabla_{\tilde{x}_i}\nabla_{\tilde{x}_j}\log q_\sigma(\tilde{x}|x)\right]
$$

$$
+ \mathbb{E}\left[s_2(\tilde{x};\theta)\nabla_{\tilde{x}_i}q_\sigma(\tilde{x}|x)\nabla_{\tilde{x}_j}q_\sigma(\tilde{x}|x)\right].
$$

**For Eq.(16):**

$$
\int s_2(\tilde{x};\theta)\frac{\nabla_{\tilde{x}_i}p(\tilde{x}) \cdot \nabla_{\tilde{x}_j}p(\tilde{x})}{p(\tilde{x})}d\tilde{x} = \int s_2(\tilde{x};\theta))\frac{\nabla_{\tilde{x}_i}p(\tilde{x})}{p(\tilde{x})}\frac{\nabla_{\tilde{x}_j}p(\tilde{x})}{p(\tilde{x})}p(\tilde{x})d\tilde{x}
$$

$$
= \mathbb{E}\left[s_2(\tilde{x};\theta)s_1^2(\tilde{x})\right].
$$

Combine all the results, we get the desired result under the MCAR mechanism. By similar argument in the proof under the MCAR mechanism, we have

$$
\mathbb{E}_{\tilde{\mathbf{x}},\mathbf{x},\mathbf{m}} \left[ \left\| \left\{ \mathbf{s}_2(\tilde{\mathbf{x}};\boldsymbol{\theta}) + \mathbf{s}_1(\tilde{\mathbf{x}})\mathbf{s}_1^\top(\tilde{\mathbf{x}}) + \frac{\mathbf{I} - \mathbf{z}\mathbf{z}^\top}{\sigma^2} \right\} \odot \frac{\left\{ (\mathbf{1}-\mathbf{m})(\mathbf{1}-\mathbf{m})^\top \right\}}{\sqrt{\mathbb{P}[\mathbf{m}\mathbf{m}^\top = 0 | \mathbf{x} = \mathbf{x}]}} \right\|_2^2 \right]
$$

$$
=\mathbb{E}_{\tilde{\mathbf{x}},\mathbf{x}} \left[ \left\| \left\{ \mathbf{s}_2(\tilde{\mathbf{x}};\boldsymbol{\theta}) + \mathbf{s}_1(\tilde{\mathbf{x}})\mathbf{s}_1^\top(\tilde{\mathbf{x}}) + \frac{\mathbf{I} - \mathbf{z}\mathbf{z}^\top}{\sigma^2} \right\} \odot \frac{\sqrt{\mathbb{E}_{\mathbf{m}|\tilde{\mathbf{x}},\mathbf{x}}(\mathbf{1}-\mathbf{m})(\mathbf{1}-\mathbf{m})^\top}}{\sqrt{\mathbb{P}[\mathbf{m}\mathbf{m}^\top = 0 | \mathbf{x} = \mathbf{x}]}} \right\|_2^2 \right]
$$

$$
=\mathbb{E}_{\tilde{\mathbf{x}},\mathbf{x}} \left[ \left\| \left\{ \mathbf{s}_2(\tilde{\mathbf{x}};\boldsymbol{\theta}) + \mathbf{s}_1(\tilde{\mathbf{x}})\mathbf{s}_1^\top(\tilde{\mathbf{x}}) + \frac{\mathbf{I} - \mathbf{z}\mathbf{z}^\top}{\sigma^2} \right\} \right\|_2^2 \right].
$$

where the last equality comes from $\mathbb{E}_{\mathbf{m}|\tilde{\mathbf{x}},\mathbf{x}}[(\mathbf{1}-\mathbf{m})(\mathbf{1}-\mathbf{m})^\top] = \mathbb{E}_{\mathbf{m}|\mathbf{x}}[(\mathbf{1}-\mathbf{m})(\mathbf{1}-\mathbf{m})^\top] = \mathbb{P}[\mathbf{m}\mathbf{m}^\top = 0 | \mathbf{x} = \mathbf{x}]$.

Then following the steps in section A.2, taking $\mathbf{w}_2$ in A.2 as 1, we get our desired result for the MAR mechanism. □

### A.3. Proof of Theorem 3.5

We first consider the MCAR case. Recall that $\mathbf{w} = \mathbf{1} - \mathbf{m}$

$$
\mathbb{E}_{\tilde{\mathbf{x}},\mathbf{x},\mathbf{m}} \left[ \| \left\{ \otimes^k \mathbf{x} - f_k(\tilde{\mathbf{x}}, \mathbf{s}_1(\tilde{\mathbf{x}}), \ldots, \mathbf{s}_{k-1}(\tilde{\mathbf{x}}), \mathbf{s}_k(\tilde{\mathbf{x}};\boldsymbol{\theta})) \right\} \odot \otimes^k \mathbf{w} \|^2 \right]
$$
$$
= \mathbb{E}_{\tilde{\mathbf{x}},\mathbf{x},\mathbf{m}} \left[ \| \left\{ \otimes^k \mathbf{x} - f_k(\tilde{\mathbf{x}}, \mathbf{s}_1(\tilde{\mathbf{x}}), \ldots, \mathbf{s}_{k-1}(\tilde{\mathbf{x}}), \mathbf{s}_k(\tilde{\mathbf{x}};\boldsymbol{\theta})) \right\} \odot \otimes^k (\mathbf{1}-\mathbf{m}) \|^2 \right]
$$
$$
= \mathbb{E}_{\tilde{\mathbf{x}},\mathbf{x}} \mathbb{E}_{\mathbf{m}|\tilde{\mathbf{x}},\mathbf{x}} \left[ \| \left\{ \otimes^k \mathbf{x} - f_k(\tilde{\mathbf{x}}, \mathbf{s}_1(\tilde{\mathbf{x}}), \ldots, \mathbf{s}_{k-1}(\tilde{\mathbf{x}}), \mathbf{s}_k(\tilde{\mathbf{x}};\boldsymbol{\theta})) \right\} \odot \otimes^k (\mathbf{1}-\mathbf{m}) \|^2 \right]
$$
$$
= \mathbb{E}_{\tilde{\mathbf{x}},\mathbf{x}} \left[ \| \left\{ \otimes^k \mathbf{x} - f_k(\tilde{\mathbf{x}}, \mathbf{s}_1(\tilde{\mathbf{x}}), \ldots, \mathbf{s}_{k-1}(\tilde{\mathbf{x}}), \mathbf{s}_k(\tilde{\mathbf{x}};\boldsymbol{\theta})) \right\} \odot \left\{ \mathbb{E}_{\mathbf{m}} \otimes^k (\mathbf{1}-\mathbf{m}) \right\} \|^2 \right]
$$

Noting that $\mathbb{E}_{\mathbf{m}} \otimes^k (\mathbf{1}-\mathbf{m})$ is a constant. We can show that the solution for the weighted least square equation for any constant matrix $\mathbf{c}$ is
$$
\arg \min_h \mathbb{E} \left[ \| \left\{ \otimes^k \mathbf{x} - h(\tilde{\mathbf{x}}) \right\} \odot \mathbf{c} \|^2 \right] = \mathbb{E} \left\{ \otimes^k \mathbf{x} \mid \tilde{\mathbf{x}} \right\}.
$$

Such equation holds since

$$
\mathbb{E} \left[ \| \left\{ \otimes^k \mathbf{x} - h(\tilde{\mathbf{x}}) \right\} \odot \mathbf{c} \|^2 \right] = \mathbb{E} \left[ \| \left\{ \otimes^k \mathbf{x} - \mathbb{E} \left[ \otimes^k \mathbf{x} \mid \tilde{\mathbf{x}} \right] \right\} \odot \mathbf{c} \|^2 \right]
$$
$$
+ \mathbb{E} \left[ \| \left\{ \mathbb{E} \left[ \otimes^k \mathbf{x} \mid \tilde{\mathbf{x}} \right] - h(\tilde{\mathbf{x}}) \right\} \odot \mathbf{c} \|^2 \right]
$$
$$
\geq \mathbb{E} \left[ \| \left\{ \otimes^k \mathbf{x} - \mathbb{E} \left[ \otimes^k \mathbf{x} \mid \tilde{\mathbf{x}} \right] \right\} \odot \mathbf{c} \|^2 \right]
$$

By Theorem 3.2, we know that $\mathbb{E} \left\{ \otimes^k \mathbf{x} \mid \tilde{\mathbf{x}} \right\} = f_k(\tilde{\mathbf{x}}, \mathbf{s}_1(\tilde{\mathbf{x}}), \ldots, \mathbf{s}_{k-1}(\tilde{\mathbf{x}}), \mathbf{s}_k(\tilde{\mathbf{x}}))$. Hence, the desired result follows.

For the MAR case, recall that $\mathbf{w} = \frac{\mathbf{1}-\mathbf{m}}{\mathbb{P}[\mathbf{m}^k = 0 | \mathbf{x} = \mathbf{x}]}$

$$
\mathbb{E}_{\tilde{\mathbf{x}},\mathbf{x},\mathbf{m}} \left[ \| \left\{ \otimes^k \mathbf{x} - f_k(\tilde{\mathbf{x}}, \mathbf{s}_1(\tilde{\mathbf{x}}), \ldots, \mathbf{s}_{k-1}(\tilde{\mathbf{x}}), \mathbf{s}_k(\tilde{\mathbf{x}};\boldsymbol{\theta})) \right\} \odot \otimes^k \mathbf{w} \|^2 \right]
$$
$$
= \mathbb{E}_{\tilde{\mathbf{x}},\mathbf{x},\mathbf{m}} \left[ \left\| \left\{ \otimes^k \mathbf{x} - f_k(\tilde{\mathbf{x}}, \mathbf{s}_1(\tilde{\mathbf{x}}), \ldots, \mathbf{s}_{k-1}(\tilde{\mathbf{x}}), \mathbf{s}_k(\tilde{\mathbf{x}};\boldsymbol{\theta})) \right\} \odot \otimes^k \frac{(\mathbf{1}-\mathbf{m})}{\mathbb{P}[\mathbf{m}^k = 0 | \mathbf{x} = \mathbf{x}]} \right\|^2 \right]
$$
$$
= \mathbb{E}_{\tilde{\mathbf{x}},\mathbf{x}} \mathbb{E}_{\mathbf{m}|\mathbf{x}} \left[ \left\| \left\{ \otimes^k \mathbf{x} - f_k(\tilde{\mathbf{x}}, \mathbf{s}_1(\tilde{\mathbf{x}}), \ldots, \mathbf{s}_{k-1}(\tilde{\mathbf{x}}), \mathbf{s}_k(\tilde{\mathbf{x}};\boldsymbol{\theta})) \right\} \odot \otimes^k \frac{(\mathbf{1}-\mathbf{m})}{\mathbb{P}[\mathbf{m}^k = 0 | \mathbf{x} = \mathbf{x}]} \right\|^2 \right]
$$
$$
= \mathbb{E}_{\tilde{\mathbf{x}},\mathbf{x}} \left[ \| \left\{ \otimes^k \mathbf{x} - f_k(\tilde{\mathbf{x}}, \mathbf{s}_1(\tilde{\mathbf{x}}), \ldots, \mathbf{s}_{k-1}(\tilde{\mathbf{x}}), \mathbf{s}_k(\tilde{\mathbf{x}};\boldsymbol{\theta})) \right\} \|^2 \right].
$$

Thus, by Theorem 3.2, the solution to the criterion function is

$$
\arg \min_{\theta} \mathbb{E}_{\tilde{\mathbf{x}},\mathbf{x},\mathbf{m}} \left[ \| \left\{ \otimes^k \mathbf{x} - f_k(\tilde{\mathbf{x}}, \mathbf{s}_1(\tilde{\mathbf{x}}), \ldots, \mathbf{s}_{k-1}(\tilde{\mathbf{x}}), \mathbf{s}_k(\tilde{\mathbf{x}};\boldsymbol{\theta})) \right\} \odot \otimes^k \mathbf{w} \|^2 \right] = \theta^*.
$$

**A.4. Proof of Eq.(7)**

Denote the oracle criterion function as

$$\widetilde{\mathcal{L}}_{\mathrm{DSM}}(\boldsymbol{\theta}) := \mathbb{E}_{\mathbf{x},\mathbf{m}}\mathbb{E}_{\tilde{\mathbf{x}}|\mathbf{x},\mathbf{m}}\left[\left\|\left\{\mathbf{s}_1(\tilde{\mathbf{x}};\boldsymbol{\theta}) + \frac{1}{\sigma^2}(\tilde{\mathbf{x}} - \mathbf{x})\right\} \odot \mathbf{g}_1(\mathbf{x},\mathbf{m})\right\|_2^2\right].$$

where $\mathbf{g}_1(\mathbf{x},\mathbf{m}) = 1 - \mathbf{m}$ under MAR and $\mathbf{g}_1(\mathbf{x},\mathbf{m}) = \frac{1-\mathbf{m}}{\sqrt{\mathbb{P}[\mathbf{m}=0|\mathbf{x}=\mathbf{x}]}}$.

If we want to match the score of true data distribution $p(\mathbf{x})$, $\sigma$ should be approximately zero for both DSM and D$_2$SM so that $q_\sigma(\tilde{\mathbf{x}})$ is close to $p(\mathbf{x})$. According to Taylor expansion we have,

$$\begin{aligned}
\widetilde{\mathcal{L}}_{\mathrm{DSM}}(\boldsymbol{\theta}) &= \mathbb{E}_{\tilde{\mathbf{x}},\mathbf{x},\mathbf{m}}\left[\left\|\left\{\mathbf{s}_1(\tilde{\mathbf{x}};\boldsymbol{\theta}) + \frac{1}{\sigma^2}(\tilde{\mathbf{x}} - \mathbf{x})\right\} \odot \mathbf{g}_1(\mathbf{x},\mathbf{m})\right\|_2^2\right] \\
&= \mathbb{E}_{\mathbf{x},\mathbf{m}}\mathbb{E}_{\mathbf{z}\sim\mathcal{N}(\mathbf{0},\mathbf{I})}\left[\left\|\left\{\mathbf{s}_1(\mathbf{x}+\sigma\mathbf{z};\boldsymbol{\theta}) + \frac{\mathbf{z}}{\sigma}\right\} \odot \mathbf{g}_1(\mathbf{x},\mathbf{m})\right\|_2^2\right] \\
&= \mathbb{E}_{\mathbf{x},\mathbf{m}}\mathbb{E}_{\mathbf{z}\sim\mathcal{N}(\mathbf{0},\mathbf{I})}\left[\left\|\left\{\mathbf{s}_1(\mathbf{x};\boldsymbol{\theta}) + \sigma\nabla_{\mathbf{x}}\mathbf{s}_1(\mathbf{x};\boldsymbol{\theta})\mathbf{z} + \frac{\mathbf{z}}{\sigma}\right\} \odot \mathbf{g}_1(\mathbf{x},\mathbf{m})\right\|_2^2\right] + \mathcal{O}(1) \\
&= \mathbb{E}_{\mathbf{x},\mathbf{m}}\mathbb{E}_{\mathbf{z}\sim\mathcal{N}(\mathbf{0},\mathbf{I})}\left[\left\|\left\{\mathbf{s}_1(\mathbf{x};\boldsymbol{\theta}) + \frac{\mathbf{z}}{\sigma}\right\} \odot \mathbf{g}_1(\mathbf{x},\mathbf{m})\right\|_2^2\right] + \mathcal{O}(1) \\
&= \mathbb{E}_{\mathbf{x},\mathbf{m}}\mathbb{E}_{\mathbf{z}\sim\mathcal{N}(\mathbf{0},\mathbf{I})}\left[\left\{\|\mathbf{s}_1(\mathbf{x};\boldsymbol{\theta}) \odot \mathbf{g}_1(\mathbf{x})\|_2^2 + \frac{\|\mathbf{z} \odot \mathbf{g}_1(\mathbf{x},\mathbf{m})\|^2}{\sigma^2} + \left(\frac{2}{\sigma}\mathbf{s}_1(\mathbf{x};\boldsymbol{\theta})^\top\mathbf{z}\right) \odot \mathbf{g}_1(\mathbf{x},\mathbf{m})\right\}\right] + \mathcal{O}(1)
\end{aligned}$$

where $\mathbf{z} = \frac{\tilde{\mathbf{x}}-\mathbf{x}}{\sigma}$, where $\mathcal{O}(1)$ is bounded as $\sigma$ approaches zero. However, when evaluating the expectation above from samples, the variances of $\frac{\|\mathbf{z} \odot \mathbf{g}(\mathbf{x},\mathbf{m})\|^2}{\sigma^2}$ and $\frac{\mathbf{s}(\mathbf{x};\boldsymbol{\theta})^\top\mathbf{z}}{\sigma}$ both increase without bound as $\sigma$ nears zero, due to the terms involving $\sigma$ and $\sigma^2$ in the denominator. This leads to a significant increase in the variance of the DSM loss, complicating the optimization process. As a consequence, DSM may become unstable and fail to converge when $\sigma$ is small, highlighting the need for methods to reduce variance.

We have,

$$\mathbb{E}_{\mathbf{z}\sim\mathcal{N}(\mathbf{0},\mathbf{I})}\left[\frac{\|\mathbf{z} \odot \mathbf{g}_1(\mathbf{x},\mathbf{m})\|^2}{\sigma^2} + \frac{2}{\sigma}\mathbf{s}_1(\mathbf{x};\boldsymbol{\theta})^\top\mathbf{z}\right] = \frac{\|\mathbf{g}_1(\mathbf{x},\mathbf{m})\|^2}{\sigma^2},$$

where $d$ is the dimension of the data distribution $p(\mathbf{x})$. Therefore, we can construct a variable that is, for sufficiently small $\sigma$, positively correlated with $\mathcal{L}_{\mathrm{DSM}}$ while having an expected value of zero:

$$c_{\boldsymbol{\theta}}(\mathbf{x};\mathbf{z}) = \left(\frac{2}{\sigma}\mathbf{s}_1(\mathbf{x};\boldsymbol{\theta})^\top\mathbf{z}\right) \odot \mathbf{g}_1(\mathbf{x},\mathbf{m}) + \frac{\|\mathbf{z} \odot \mathbf{g}_1(\mathbf{x},\mathbf{m})\|^2}{\sigma^2} - \frac{\|\mathbf{g}_1(\mathbf{x},\mathbf{m})\|^2}{\sigma^2}.$$

Subtracting it from $\mathcal{L}_{\mathrm{DSM}}$ will yield an estimator with reduced variance for DSM training with missing data:

$$\mathcal{L}_{\mathrm{DSM\text{-}VR}}(\boldsymbol{\theta}) = \mathcal{L}_{\mathrm{DSM}}(\boldsymbol{\theta}) - \mathbb{E}_{\mathbf{x},\mathbf{m}}\mathbb{E}_{\mathbf{z}\sim\mathcal{N}(\mathbf{0},\mathbf{I})}\left[\left(\frac{2}{\sigma}\mathbf{s}(\mathbf{x};\boldsymbol{\theta})^\top\mathbf{z}\right) \odot \mathbf{g}_1(\mathbf{x},\mathbf{m}) + \frac{\|\mathbf{z} \odot \mathbf{g}_1(\mathbf{x},\mathbf{m})\|^2}{\sigma^2}\right].$$

Here we omit the part $\frac{\|\mathbf{g}_1(\mathbf{x},\mathbf{m})\|^2}{\sigma^2}$ since it is independent of $\boldsymbol{\theta}$.

**A.5. Proof of Eq.(8)**

Similar to proof A.4, consider the oracle criterion function

$$\widetilde{L}_{\mathrm{D}_2\mathrm{SM}}(\boldsymbol{\theta}) = \mathbb{E}_{\tilde{\mathbf{x}},\mathbf{x},\mathbf{m}}\left[\left\|\left\{\mathbf{s}_2(\tilde{\mathbf{x}};\boldsymbol{\theta}) + \mathbf{s}_1(\tilde{\mathbf{x}})\mathbf{s}_1^\top(\tilde{\mathbf{x}}) + \frac{\mathbf{I} - \mathbf{z}\mathbf{z}^\top}{\sigma^2}\right\} \odot \mathbf{g}_2(\mathbf{x},\mathbf{m})\right\|_2^2\right],$$

where $\mathbf{g}_2(\mathbf{x}_{\text{obs}}, \mathbf{m}) = (\mathbf{1} - \mathbf{m})(\mathbf{1} - \mathbf{m})^\top$ under MCAR, and $\mathbf{g}_2(\mathbf{x}_{\text{obs}}, \mathbf{m}) = \frac{(\mathbf{1}-\mathbf{m})(\mathbf{1}-\mathbf{m})^\top}{\sqrt{\mathbb{E}[\mathbf{m}\mathbf{m}^\top | \mathbf{x}=\mathbf{x}_{\text{obs}}]}}$ under MAR.

Denote $\boldsymbol{\psi}(\tilde{\mathbf{x}}; \boldsymbol{\theta}) = \mathbf{s}_2(\tilde{\mathbf{x}}_{\text{obs}}; \boldsymbol{\theta}) + \mathbf{s}_1(\tilde{\mathbf{x}}_{\text{obs}})\mathbf{s}_1^\top(\tilde{\mathbf{x}}_{\text{obs}})$, we have

$$
\begin{aligned}
\widetilde{\mathcal{L}}_{\text{D}_2\text{SM}}(\boldsymbol{\theta}) &= \mathbb{E}_{\tilde{\mathbf{x}}, \mathbf{x}, \mathbf{m}}\left[\left\|\left\{\mathbf{s}_2(\tilde{\mathbf{x}}; \boldsymbol{\theta}) + \mathbf{s}_1(\tilde{\mathbf{x}})\mathbf{s}_1^\top(\tilde{\mathbf{x}}) + \frac{\mathbf{I} - \mathbf{z}\mathbf{z}^\top}{\sigma^2}\right\} \odot \mathbf{g}_2(\mathbf{x}, \mathbf{m})\right\|_2^2\right] \\
&= \mathbb{E}_{\mathbf{x}, \mathbf{m}}\mathbb{E}_{\mathbf{z} \sim \mathcal{N}(\mathbf{0}, \mathbf{I})}\left[\left\|\left\{\boldsymbol{\psi}(\mathbf{x} + \sigma\mathbf{z}; \boldsymbol{\theta}) + \frac{\mathbf{I} - \mathbf{z}\mathbf{z}^\top}{\sigma^2}\right\} \odot \mathbf{g}_2(\mathbf{x}, \mathbf{m})\right\|_2^2\right] \\
&= \mathbb{E}_{\mathbf{x}, \mathbf{m}}\mathbb{E}_{\mathbf{z} \sim \mathcal{N}(\mathbf{0}, \mathbf{I})}\left[\left\{\|\boldsymbol{\psi}(\mathbf{x} + \sigma\mathbf{z}; \boldsymbol{\theta})\|_2^2 + \left\|\frac{\mathbf{I} - \mathbf{z}\mathbf{z}^\top}{\sigma^2}\right\|_2^2 + 2\boldsymbol{\psi}(\mathbf{x} + \sigma\mathbf{z}; \boldsymbol{\theta})\frac{\mathbf{I} - \mathbf{z}\mathbf{z}^\top}{\sigma^2}\right\} \odot \mathbf{g}_2(\mathbf{x}, \mathbf{m})\right]
\end{aligned}
$$

Denote $\psi_{ij}(\tilde{x}; \theta)$ as the $ij$th term of $\boldsymbol{\psi}(\tilde{\mathbf{x}}; \boldsymbol{\theta})$, $\phi_{ij} = \mathbf{I}_{ij} - \mathbf{z}_i\mathbf{z}_j$ and $g_{ij}$ as the $ij$th term of $\mathbf{g}_2(\mathbf{x}, \mathbf{m})$ and according to Taylor expansion, we have,

$$
\begin{aligned}
&\mathbb{E}_x\mathbb{E}_{z \sim \mathcal{N}(0, I)}\left[\left\{\psi_{ij}(x + \sigma z; \theta)^2 + \frac{\phi_{ij}}{\sigma^2} + 2\psi_{ij}(x + \sigma z; \theta)\frac{\phi_{ij}}{\sigma^2}\right\} \odot g_{ij}(x, m)\right] \\
&= \mathbb{E}_x\mathbb{E}_{z \sim \mathcal{N}(0, I)}\left[\left\{\psi_{ij}(x; \theta)^2 + 2\psi_{ij}(x; \theta)\frac{\phi_{ij}}{\sigma^2} + 2\nabla\psi_{ij}(x; \theta)\frac{\phi_{ij}}{\sigma} + C\right\} \odot g_{ij}(x, m)\right] + \mathcal{O}(1)
\end{aligned}
$$

where $z = \frac{\tilde{x}-x}{\sigma}$, with $C = \left(\frac{\phi_{ij}}{\sigma^2}\right)^2$ and $\nabla\psi_{ij}(x + \sigma z; \theta)$ representing the derivative of $\psi_{ij}(x + \sigma z; \theta)$ with respect to $x$. $\left(\frac{\phi_{ij}}{\sigma^2}\right)^2$ can be treated as a constant that does not depend on $\boldsymbol{\theta}$, and $\mathcal{O}(1)$ remains bounded when $\sigma \to 0$. However, when calculating the expectation from samples, the variances of $\frac{\phi_{ij}}{\sigma^2}$ and $\nabla\psi_{ij}(x; \theta)\frac{\phi_{ij}}{\sigma^2}$ increase without bound as $\sigma \to 0$, due to the presence of $\sigma$ and $\sigma^2$ in the denominator. This causes a significant rise in variance for D$_2$SM, making the optimization process more difficult. Consequently, D$_2$SM can become unstable and fail to converge as $\sigma$ approaches zero, necessitating the use of variance reduction techniques.

In this case, we can employ the same variance reduction method outlined in the proof of A.4. However, to bypass the need for estimating $\nabla\psi_{ij}(x; \theta)\frac{\phi_{ij}}{\sigma^2}$, we utilize the antithetic sampling technique same as Meng et al. (2021) to reduce variance.

Denote $\tilde{x}_+ = x + \sigma z$ and $\tilde{x}_- = x + \sigma z$ as the antithetic samples, according to Taylor expansion, the $ij$th term of the $D_2\text{SM}(\theta)$ then becomes,

$$
\begin{aligned}
\widetilde{\mathcal{L}}_{D_2\text{SM}}(\theta)_{ij} &= \mathbb{E}_x\mathbb{E}_{z \sim \mathcal{N}(0, I)}\left[\left\{\left(\psi_{ij}(\tilde{x}_+; \theta) + \frac{\phi_{ij}}{\sigma^2}\right)^2 + \left(\psi_{ij}(\tilde{x}_-; \theta) + \frac{\phi_{ij}}{\sigma^2}\right)^2\right\} \odot g_{ij}(x, m)\right] \\
&= \mathbb{E}_x\mathbb{E}_{z \sim \mathcal{N}(0, I)}\left[\left\{\psi_{ij}(\tilde{x}_+; \theta)^2 + 2\frac{\phi_{ij}}{\sigma^2}\psi_{ij}(\tilde{x}_+; \theta) + \psi_{ij}(\tilde{x}_-; \theta)^2 + 2\frac{\phi_{ij}}{\sigma^2}\psi_{ij}(\tilde{x}_-; \theta) + C\right\} \odot g_{ij}(x, m)\right] \\
&= \mathbb{E}_x\mathbb{E}_{z \sim \mathcal{N}(0, I)}\left[\left\{\psi_{ij}(\tilde{x}_+; \theta)^2 + \psi_{ij}(\tilde{x}_-; \theta)^2 + 2\frac{\phi_{ij}}{\sigma^2}\left(\psi_{ij}(\tilde{x}_+; \theta) + \psi_{ij}(\tilde{x}_-; \theta)\right) + C\right\} \odot g_{ij}(x, m)\right] \\
&= \mathbb{E}_x\mathbb{E}_{z \sim \mathcal{N}(0, I)}\left[\left\{2\left(\psi_{ij}(x; \theta)\right)^2 + \frac{\phi_{ij}}{\sigma^2}\left[4\psi_{ij}(x; \theta) + 2\nabla\psi_{ij}(x; \theta) - 2\nabla\psi_{ij}(x; \theta)\right] + C\right\} \odot g_{ij}(x, m)\right] \\
&= \mathbb{E}_x\mathbb{E}_{\mathbf{z} \sim \mathcal{N}(\mathbf{0}, \mathbf{I})}\left[\left\{2\left(\psi_{ij}(x; \theta)\right)^2 + 4\frac{\phi_{ij}}{\sigma^2}\left(\psi_{ij}(x; \theta)\right) + C\right\} \odot g_{ij}(x, m)\right],
\end{aligned}
$$

where $C = 2\left(\frac{\phi_{ij}}{\sigma^2}\right)^2$, a constant with respect to the optimization.

Therefore, we have the variance reduction for $\mathcal{L}_{\text{D}_2\text{SM-VR}}(\boldsymbol{\theta})$ which is equivalent to optimizing Eq. (3.4) up to a control variate. Moreover, when $\sigma$ approaches zero, optimizing Eq. (8) is more stable.

$$
\mathcal{L}_{\text{D}_2\text{SM-VR}}(\boldsymbol{\theta}) = \mathbb{E}_{\mathbf{x}, \mathbf{m}}\mathbb{E}_{\mathbf{z} \sim \mathcal{N}(\mathbf{0}, \mathbf{I})}\left[\left\{\boldsymbol{\psi}(\tilde{\mathbf{x}}^+)^2 + \boldsymbol{\psi}(\tilde{\mathbf{x}}^-)^2 + 2\frac{\mathbf{I} - \mathbf{z}\mathbf{z}^\top}{\sigma} \odot \boldsymbol{\Psi}(\cdot)\right\} \odot \mathbf{g}_2(\mathbf{x}, \mathbf{m})\right], \tag{17}
$$

---

**Algorithm 1** MissScore

---

**Input:** Observed data $\mathbf{x}_{\text{obs}}$, score models $\mathbf{s}_1(\cdot; \boldsymbol{\theta})$, $\mathbf{s}_2(\cdot; \boldsymbol{\theta})$, noise level $\sigma$, coefficient $\omega$
Infer the missingness mask $\mathbf{m} = \mathbf{1}_{[\mathbf{x}_{\text{obs}}=\text{na}]}$
**repeat**
    Sample noise $\mathbf{z} \sim \mathcal{N}(\mathbf{0}, \mathbf{I})$
    Compute perturbed data: $\tilde{\mathbf{x}}_{\text{obs}} = \mathbf{x}_{\text{obs}} + \sigma\mathbf{z}$
    **if** missing mechanism is MAR **then**
        Estimate $\mathbb{P}[\mathbf{m} = 0 | \mathbf{x} = \mathbf{x}_{\text{obs}}]$ and $\mathbb{P}[\mathbf{m}\mathbf{m}^\top = 0 | \mathbf{x} = \mathbf{x}_{\text{obs}}]$ using fitted logistic models
    **end if**
    Update parameters using gradient descent on $\nabla_{\boldsymbol{\theta}}\big(\text{Eq.}(4) + \omega\,\text{Eq.}(5)\big)$
**until** convergence

---

where the antithetic samples are defined as $\mathbf{x}^+ = \mathbf{x} + \sigma\mathbf{z}$ and $\mathbf{x}^- = \mathbf{x} - \sigma\mathbf{z}$. Here, $\boldsymbol{\psi} = \mathbf{s}_2 + \mathbf{s}_1\mathbf{s}_1^\top$, and $\boldsymbol{\Psi} = \big(\boldsymbol{\psi}(\tilde{\mathbf{x}}^+) + \boldsymbol{\psi}(\tilde{\mathbf{x}}^-) - 2\boldsymbol{\psi}(\mathbf{x})\big)$.

## B. Data Generation under Different Missing Mechanisms

Missing data mechanisms can vary significantly, but they are typically categorized into three main types as defined by Rubin (1976): missing completely at random (MCAR), missing at random (MAR), and missing not at random (MNAR). In our experiments, we simulate missing data based on these mechanisms as follows:

**MCAR (Missing Completely at Random)**: Missing values are generated uniformly, with each data point having an equal probability of being missing, determined by a predefined missing rate, $\alpha$. Specifically, missing values are generated using a Bernoulli distribution, $\text{Ber}(\alpha)$, where each entry is missing independently with probability $\alpha$.

**MAR (Missing at Random)**: In this scenario, missing values are generated using a logistic model. A random subset of the variables is selected to remain fully observed, while the remaining variables have missing values depending on the fully observed ones. The missingness is determined by a logistic model with random coefficients, scaled to achieve the target proportion of missing data for the variables influenced by the fully observed subset.

**MNAR (Missing Not at Random)**: The MNAR mechanism is modeled using a logistic masking model. It implements two mechanisms and in either case, weights are random and the intercept is selected to attain the desired proportion of missing values.

- Missing probabilities for each variable are determined by a logistic model that takes all the variables (including those with missing data) as inputs;

- Variables are split into two sets: a set of input variables for the logistic model and a set of variables whose missingness is determined by the logistic model. The input variables are masked using an MCAR process, meaning the missingness in the second set depends on the missingness in the input set.

In all experiments, for MAR missing mechanism, we use logistic regression to estimate the likelihood of each data point being observed. For MNAR, we utilize the same training objective as MCAR, while recognizing that this method may introduce some bias. The algorithm for training the first- and second-order models under different missing mechanisms is outlined in Algorithm 1.

**Scalability and numerical stability for MAR**    We have included additional synthetic results for the MAR mechanism in Table 3, following the same setup as described in Section 4.2. The results exhibit a similar pattern, though slightly worse than MCAR, which may be attributed to the inverse probability, potentially increasing variance and leading to less stable estimations. However, the estimates remain reasonable and demonstrate empirical performance closely aligned with the ground truth, along with variance reduction. Furthermore, to illustrate the impact of potential model misspecification by the logistic regression model in MAR, we conduct an experiment comparing ground truth $p$ with the estimated $\hat{p}$ from the logistic regression model on the same synthetic dataset, with varying missing ratios. When comparing the performance shown in Tables 3 and 4, although potential model misspecification in the logistic regression model influences the estimation of missing probabilities, this impact is kept within a reasonable range in the variance-reduced version.

---

**Algorithm 2** MissScore for Sampling with Missing Data

---

**Input:** Score models $\mathbf{s}_1(\cdot; \boldsymbol{\theta}^*)$, $\mathbf{s}_2(\cdot; \boldsymbol{\theta}^*)$; step size $\epsilon$; number of iterations $T$
**Initialize:** $\tilde{\mathbf{x}}_0 \sim \pi(\mathbf{x})$ $t = 1$ to $T$
Sample noise $\mathbf{z}_t \sim \mathcal{N}(\mathbf{0}, \mathbf{I})$
**if** Ozaki sampling **then**
  Compute $\mathbf{M}_{t-1} = \left(e^{\epsilon \mathbf{s}_2(\tilde{\mathbf{x}}_{t-1})} - \mathbf{I}\right) \mathbf{s}_2(\tilde{\mathbf{x}}_{t-1})^{-1}$
  Compute $\Sigma_{t-1} = \left(e^{2\epsilon \tilde{\mathbf{x}}_{t-1}} - \mathbf{I}\right) \mathbf{s}_2(\tilde{\mathbf{x}}_{t-1})^{-1}$
  Update $\tilde{\mathbf{x}}_t = \tilde{\mathbf{x}}_{t-1} + \mathbf{M}_{t-1} \mathbf{s}_1(\tilde{\mathbf{x}}_{t-1}) + \Sigma_{t-1}^{1/2} \mathbf{z}_t$
**else**
  Update $\tilde{\mathbf{x}}_t = \tilde{\mathbf{x}}_{t-1} + \frac{1}{2}\epsilon \mathbf{s}_1(\tilde{\mathbf{x}}_{t-1}) + \sqrt{\epsilon}\mathbf{z}_t$
**end if**
**Return:** $\tilde{\mathbf{x}}_T$

---

*Table 3.* Mean squared error (MSE) between the estimated first-order and second-order scores and the ground truth is evaluated across 5,000 test samples. We vary the noise scales $\sigma$ and missing ratios $\alpha$, with each configuration tested using 10 random seeds. **MAR with estimated missing probability $\hat{p}$ by logistic model.**

| Methods | $\alpha = 0.0$ (Complete data) | | $\alpha = 0.1$ | | $\alpha = 0.3$ | | $\alpha = 0.5$ | |
|---|---|---|---|---|---|---|---|---|
| | $\sigma = 0.1$ | $\sigma = 0.01$ | $\sigma = 0.1$ | $\sigma = 0.01$ | $\sigma = 0.1$ | $\sigma = 0.01$ | $\sigma = 0.1$ | $\sigma = 0.01$ |
| $\mathbf{s}_1$ | $0.28 \pm 0.01$ | $0.42 \pm 0.02$ | $0.32 \pm 0.00$ | $0.51 \pm 0.02$ | $0.36 \pm 0.02$ | $0.52 \pm 0.00$ | $0.45 \pm 0.01$ | $0.56 \pm 0.03$ |
| $\mathbf{s}_1$(VR) | $0.07 \pm 0.00$ | $0.07 \pm 0.00$ | $0.11 \pm 0.02$ | $0.13 \pm 0.01$ | $0.15 \pm 0.01$ | $0.16 \pm 0.02$ | $0.22 \pm 0.02$ | $0.24 \pm 0.04$ |
| $\mathbf{s}_2$ | $0.16 \pm 0.02$ | $15.42 \pm 0.47$ | $0.28 \pm 0.02$ | $31.22 \pm 1.08$ | $0.30 \pm 0.04$ | $34.24 \pm 5.14$ | $0.36 \pm 0.04$ | $35.41 \pm 1.03$ |
| $\mathbf{s}_2$(VR) | $0.04 \pm 0.00$ | $0.05 \pm 0.00$ | $0.04 \pm 0.00$ | $0.05 \pm 0.00$ | $0.06 \pm 0.00$ | $0.06 \pm 0.00$ | $0.08 \pm 0.01$ | $0.07 \pm 0.00$ |

*Table 4.* Mean squared error (MSE) between the estimated first-order and second-order scores and the ground truth is evaluated across 5,000 test samples. We vary the noise scales $\sigma$ and missing ratios $\alpha$, with each configuration tested using 10 random seeds. **MAR with ground truth missing probability $p$.**

| Methods | $\alpha = 0.0$ (Complete data) | | $\alpha = 0.1$ | | $\alpha = 0.3$ | | $\alpha = 0.5$ | |
|---|---|---|---|---|---|---|---|---|
| | $\sigma = 0.1$ | $\sigma = 0.01$ | $\sigma = 0.1$ | $\sigma = 0.01$ | $\sigma = 0.1$ | $\sigma = 0.01$ | $\sigma = 0.1$ | $\sigma = 0.01$ |
| $\mathbf{s}_1$ | $0.28 \pm 0.01$ | $0.42 \pm 0.02$ | $0.24 \pm 0.01$ | $0.42 \pm 0.02$ | $0.27 \pm 0.01$ | $0.41 \pm 0.01$ | $0.33 \pm 0.01$ | $0.43 \pm 0.04$ |
| $\mathbf{s}_1$(VR) | $0.07 \pm 0.00$ | $0.07 \pm 0.00$ | $0.06 \pm 0.00$ | $0.06 \pm 0.00$ | $0.09 \pm 0.00$ | $0.10 \pm 0.00$ | $0.19 \pm 0.00$ | $0.22 \pm 0.01$ |
| $\mathbf{s}_2$ | $0.16 \pm 0.02$ | $15.42 \pm 0.47$ | $0.24 \pm 0.01$ | $16.66 \pm 4.34$ | $0.27 \pm 0.01$ | $41.47 \pm 7.28$ | $0.33 \pm 0.01$ | $39.89 \pm 4.03$ |
| $\mathbf{s}_2$(VR) | $0.04 \pm 0.00$ | $0.05 \pm 0.00$ | $0.02 \pm 0.00$ | $0.03 \pm 0.00$ | $0.03 \pm 0.00$ | $0.03 \pm 0.00$ | $0.05 \pm 0.00$ | $0.05 \pm 0.00$ |

## C. Additional Information on Sampling

In the sampling experiments with the Swiss-Roll dataset under MCAR, we use a small perturbation $\sigma = 0.01$ and jointly optimize Eq. (7) and Eq.(8), where $\mathbf{s}_1(\tilde{\mathbf{x}}) \approx \mathbf{s}_1(\mathbf{x})$ and $\mathbf{s}_2(\tilde{\mathbf{x}}) \approx \mathbf{s}_2(\mathbf{x})$. The sample size is set to 5000. Both $\mathbf{s}_1(\tilde{\mathbf{x}}; \boldsymbol{\theta})$ and $\mathbf{s}_2(\tilde{\mathbf{x}}; \boldsymbol{\theta})$ are modeled using a 3-layer MLP with a latent size of 128 and a Softplus activation function. We use a learning rate of 0.001, a batch size of 64 and train for 100 epochs, which takes approximately 4 minutes on an Intel(R) Xeon(R) Gold 6448H CPU. The experiments in Section 4 also utilize the same model configuration for training. In the Ozaki sampling experiments, we only use the diagonal of $\mathbf{s}_2(\tilde{\mathbf{x}}; \boldsymbol{\theta})$ to avoid the computational costs associated with the inversion, exponentiation, and decomposition of $\mathbf{s}_2(\tilde{\mathbf{x}}; \boldsymbol{\theta})$. The algorithm is presented in Algorithm 2.

The following sections provide experimental details on data generation with the simulated Bayesian Network and real Census data.

### C.1. Dataset Description and Processing

**Bayesian Network** Details regarding the data generated from a Bayesian Network can be found in Section B.1 in (Ouyang et al., 2023).

**Census** Census dataset is a binary classification dataset that predict whether income exceeds 50K/yr based on census data (Kohavi, 1996). Also known as Adult dataset.

The statistical information of datasets used in our experiments is in Table 5. #train, #test, #continuous, and #categorical mean the number of training data, testing data, continuous columns, and categorical columns, respectively.

*Table 5.* Synthetic and Real-World Datasets Used in Experiments.

| Dataset | #Train | #Test | #Categorical | #Continuous |
|---|---|---|---|---|
| Bayesian Network | 2000 | 20000 | 3 | 2 |
| Census | 16000 | 4000 | 9 | 6 |

For data processing, we follow standard pre- and post-processing procedures for mixed-type tabular data. Specifically, we apply min-max normalization to continuous variables and reverse this scaling during generation. For discrete variables, we use one-hot encoding and apply a rounding function after the softmax function during generation.

### C.2. Evaluation Methods

We adopt the "train on synthetic, test on real (TSTR)" framework (Esteban et al., 2017), a widely used method for assessing the quality of sampling data from generative model (Kim et al., 2022; Ouyang et al., 2023; Li et al., 2019). The experimental results for sampling in this paper are calculated as follows:

1. We first download a dataset and use its existing train-test split.

2. Then we generate synthetic records equal in number to the original training set using various synthetic data generation methods.

3. Using the synthetic training records from Step 2, we train base classifiers to make predictions. We conduct a hyperparameter search for each classifier, considering Decision Tree, AdaBoost, Logistic Regression, MLP Classifiers, Random Forest, and XGBoost for the classification tasks. The hyperparameters and their candidate settings follow those described in (Kim et al., 2022; Ouyang et al., 2023), and are summarized in Table 26 of (Kim et al., 2022).

4. Finally, we evaluate the classifiers using the testing dataset, applying a range of evaluation metrics for comprehensive assessment.

Steps 2 to 4 are repeated three times for each dataset, and the average scores for each method across all evaluation metrics are calculated. The detailed metrics used in our experiment include:

1. **Accuracy**: This is calculated using the `accuracy_score` function from the `sklearn.metrics` module.

2. **Weighted-F1**:

$$\text{Weighted-F1} = \sum_{i=0}^{N} w_i s_i$$

where $N$ is the total number of classes. The weight for the $i$-th class, $w_i = \frac{1-p_i}{N-1}$, with $p_i$ representing the proportion of the $i$-th class's size relative to the total dataset. Here, $s_i$ is the F1 score for the $i$-th class, calculated using the One-vs-Rest strategy. This weighting approach is designed to prioritize the evaluation of synthesized tables by giving more importance to smaller classes, which are often prone to being overlooked by the model, thus addressing mode collapse.

3. **AUROC**: This is calculated using the `roc_auc_score` function from the `sklearn.metrics` module.

4. **SDMetrics**: This metric evaluates synthetic data by comparing it against the real data, as described in (Dat, 2023).

Among all metrics, a higher score indicates better overall quality of the synthetic data.

## C.3. Model Architecture

We use a perturbation of $\sigma = 0.1$ and jointly optimize Eq. (7) and Eq. (8). In the Bayesian Network experiment, we follow the same configuration as described earlier, with each run taking approximately 20 minutes. For the census dataset, we employ a simple MLP consisting of 5 Linear layers, LeakyReLU activation, Layer Normalization, and Dropout with a probability of 0.2 in the first layer. The learning rate is set to 0.001. The first two layers use a latent size of 128, while the last three layers use a latent size of 1024. We train with a batch size of 256 for 250 epochs, with each experiment taking approximately 4 hours. All experiments are performed on an Intel(R) Xeon(R) Gold 6448H CPU. For the downstream classifier, we use the same base hyperparameters as listed in Table 26 of (Kim et al., 2022).

## C.4. Experimental Results

In the following experimental results, we use a missing data ratio of $\alpha = 0.3$ and apply XGBoost for the downstream tasks, without delving into specific implementation details. Table 6 presents the utility evaluation of MissScore using both Langevin and Ozaki samplings, compared to other baseline methods, on the Census dataset with a missing ratio of 0.3. Additionally, Table 7 summarizes the Accuracy, AUROC, and Weighted-F1 metrics as the missing ratio varies. Figure 5 illustrates the fidelity evaluation of MissScore, again using Langevin and Ozaki samplings alongside other baselines, on the Bayesian dataset.

*Table 6.* Utility evaluation of MissScore using Langevin and Ozaki samplings, along with other baselines, on the Census dataset with missing ratio 0.3.

| Criterion | Mechanism | Langevin | Ozaki | DSM-delete | DSM-mean | STaSy-mean |
|-----------|-----------|----------|-------|-----------|----------|-----------|
| Accuracy | MCAR | 0.80 | **0.81** | 0.70 | 0.75 | 0.77 |
| | MAR | **0.82** | **0.82** | 0.69 | 0.77 | 0.74 |
| | MNAR | **0.81** | 0.80 | 0.59 | 0.80 | 0.75 |
| AUROC | MCAR | **0.84** | 0.84 | 0.57 | 0.67 | 0.62 |
| | MAR | 0.85 | **0.86** | 0.46 | 0.75 | 0.61 |
| | MNAR | **0.86** | **0.86** | 0.52 | 0.76 | 0.63 |
| Weighted-F1 | MCAR | **0.52** | **0.52** | 0.24 | 0.32 | 0.41 |
| | MAR | **0.61** | 0.60 | 0.32 | 0.38 | 0.38 |
| | MNAR | **0.69** | 0.68 | 0.41 | 0.52 | 0.42 |

*Table 7.* Evaluation of MissScore using Ozaki samplings on the Census dataset with varying missing ratios $\{0.1, 0.3, 0.5, 0.7, 0.9\}$.

| Methods | | $\alpha = 0.1$ | $\alpha = 0.3$ | $\alpha = 0.5$ | $\alpha = 0.7$ | $\alpha = 0.9$ |
|---------|------|-----|-----|-----|-----|-----|
| Accuracy | MCAR | 0.81 | 0.81 | 0.80 | 0.79 | 0.77 |
| | MAR | 0.71 | 0.82 | 0.82 | 0.82 | 0.79 |
| | MNAR | 0.80 | 0.80 | 0.83 | 0.74 | 0.72 |
| AUROC | MCAR | 0.85 | 0.84 | 0.84 | 0.86 | 0.61 |
| | MAR | 0.85 | 0.86 | 0.87 | 0.85 | 0.83 |
| | MNAR | 0.85 | 0.87 | 0.86 | 0.85 | 0.80 |
| Weighted-F1 | MCAR | 0.54 | 0.52 | 0.41 | 0.66 | 0.22 |
| | MAR | 0.64 | 0.60 | 0.67 | 0.65 | 0.63 |
| | MNAR | 0.46 | 0.68 | 0.61 | 0.64 | 0.63 |

## C.5. Broader Empirical Validation

To further assess the robustness and generalizability of MissScore, we conduct a comprehensive empirical evaluation on 27 real-world tabular datasets covering both classification and regression tasks. The datasets, sourced from the UCI Machine Learning Repository and scikit-learn, encompass a wide range of characteristics, including varying sample sizes, feature dimensions, and class distributions. Following the experimental setup established in ForestDiffusion (Jolicoeur-Martineau

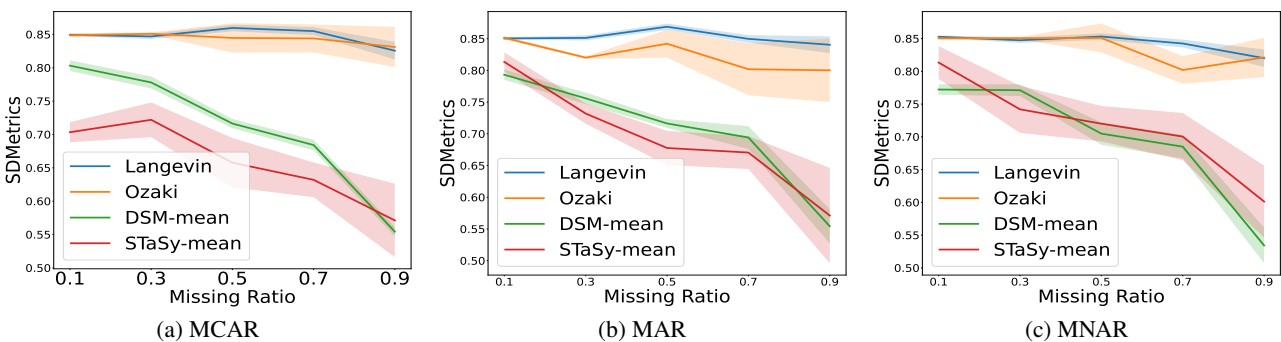

*Figure 5.* Fidelity evaluation of MissScore using Langevin and Ozaki samplings, along with other baselines, on the Census dataset varies with missing ratio $\alpha = \{0.1, 0.3, 0.5, 0.7\}$ under different missing mechanisms.

et al., 2024), we introduce 20% missingness and benchmark MissScore against a diverse set of generative baselines including copula-based models (GaussianCopula), VAE-based approaches (TVAE), score-based methods (STaSy, TabDDPM), and diffusion- or flow-based architectures (Forest-VP, Forest-Flow). Full details on dataset statistics, evaluation metrics, and baseline configurations can be found in Appendices B.1 through B.3 in ForestDiffusion (Jolicoeur-Martineau et al., 2024).

*Table 8.* Tabular data generation with incomplete data (27 datasets, 3 runs each, 20% missing); values are mean (standard-error).

| Method | $W_{train}$ ↓ | $W_{test}$ ↓ | $cov_{train}$ ↑ | $cov_{test}$ ↑ | $R^2_{fake}$ ↑ | $F1_{fake}$ ↑ | $F1_{disc}$ ↓ | $P_{bias}$ ↓ | $cov_{rate}$ ↑ |
|---|---|---|---|---|---|---|---|---|---|
| GaussianCopula | 2.60 (0.58) | 2.86 (0.63) | 0.20 (0.04) | 0.43 (0.05) | 0.20 (0.18) | 0.48 (0.06) | 0.60 (0.04) | 2.31 (1.00) | 0.21 (0.08) |
| TVAE | 2.17 (0.60) | 2.40 (0.65) | 0.32 (0.04) | 0.63 (0.04) | -0.66 (0.95) | 0.55 (0.08) | 0.45 (0.01) | 4.04 (2.30) | 0.29 (0.09) |
| Stasy | 3.40 (1.37) | 3.67 (1.40) | 0.38 (0.05) | 0.63 (0.06) | 0.27 (0.28) | 0.64 (0.06) | 0.46 (0.02) | **1.09** (0.22) | 0.36 (0.10) |
| TabDDPM | 4.36 (1.89) | 4.80 (1.90) | **0.72** (0.06) | 0.71 (0.06) | **0.58** (0.11) | 0.61 (0.10) | **0.42** (0.02) | 1.16 (0.35) | **0.56** (0.10) |
| Forest-VP | 1.84 (0.51) | 2.14 (0.56) | 0.53 (0.04) | 0.78 (0.03) | 0.53 (0.10) | **0.71** (0.04) | **0.42** (0.01) | 1.16 (0.30) | 0.43 (0.12) |
| Forest-Flow | 1.82 (0.51) | 2.12 (0.56) | 0.67 (0.03) | **0.84** (0.03) | 0.55 (0.11) | 0.69 (0.04) | 0.43 (0.01) | 1.16 (0.32) | 0.50 (0.10) |
| MissScore-Langevin | 1.85 (0.52) | 2.15 (0.57) | 0.65 (0.04) | 0.82 (0.03) | 0.54 (0.11) | 0.68 (0.04) | 0.44 (0.02) | 1.18 (0.33) | 0.49 (0.11) |
| MissScore-Ozaki | **1.79** (0.50) | **2.11** (0.55) | 0.70 (0.03) | 0.83 (0.03) | **0.58** (0.10) | 0.70 (0.04) | 0.43 (0.02) | 1.15 (0.30) | 0.52 (0.10) |

The full quantitative results are presented in Table 8. MissScore-Ozaki achieves the lowest Wasserstein distance on both training and test distributions, indicating that the generated samples closely match the real data in distributional geometry. It also performs competitively on downstream metrics such as $R^2_{fake}$ and $F1_{fake}$, which measure how well synthetic data supports regression and classification tasks. Across most metrics, both MissScore-Langevin and MissScore-Ozaki demonstrate performance comparable to or better than strong baselines, including TabDDPM and Forest-Flow, while maintaining robustness under missing data.

While MissScore does not outperform all baselines across every individual metric, we observe that it excels in several recurring settings. First, on datasets with well-separated class structures and moderate sample sizes (e.g., *seeds*), MissScore is able to model the score field accurately and generate high-quality data that supports effective downstream classification. Second, on datasets with balanced but complex class relationships (e.g., *parkinsons*), it captures nuanced inter-feature dependencies better than methods relying on shallow generative priors. Finally, in moderate-to-high dimensional datasets (e.g., *qsar biodegradation*), the use of second-order score matching enables MissScore to model non-linear feature interactions more effectively than first-order methods.

However, the method's limitations emerge in data-scarce regimes. On very small datasets like *concrete slump* (with only 103 samples), high model flexibility can induce overfitting, undermining score stability. Similarly, for datasets with many classes and few samples per class, such as *libras*, the variance in score estimation grows substantially, leading to degraded generation quality. These limitations point to directions for future work, including lightweight score architectures and adaptive regularization under low-data conditions. Overall, this extensive evaluation suggests that MissScore offers a strong and principled approach to learning from incomplete tabular data, especially in regimes where capturing higher-order structure is crucial.

# D. Additional Information on Causal Discovery

## D.1. Related Work

**Causal Discovery with Complete data.** Causal discovery aims to uncover the underlying causal relationships among variables of interest from purely observational data, specifically identifying a causal Directed Acyclic Graph (DAG) for a given dataset. This problem lies at the heart of causal inference, as knowledge of the causal graph enables prediction of the effects of interventions. However, causal discovery from observational data is inherently ill-posed, necessitating additional assumptions, such as imposing functional assumptions on the data-generating process. We adopt the notion of structural causal model (SCM) to characterize the causal relations among variables. Each SCM $\mathcal{M} = \langle \mathcal{Z}, \mathcal{X}, \mathcal{F} \rangle$ consists of the exogenous variable set $\mathcal{Z} = \{Z_1, Z_2, \ldots, Z_d\}$, the endogenous variable set $\mathcal{X} = \{X_1, X_2, \ldots, X_d\}$, and the function set $\mathcal{F} = \{f_1, f_2, \ldots, f_d\}$. Here, each function $f_i$ computes the variable $X_i$ from its parents (or causes) $X_{\mathrm{PA}_i}$ and an exogenous variable $Z_i$, i.e., $X_i = f_i(X_{\mathrm{PA}_i}, Z_i)$. We focus on a specific class of SCMs, called the additive noise models (ANMs), given by $X_i = f_i(X_{\mathrm{PA}_i}) + Z_i, \quad i = 1, 2, \ldots, d$, where $Z_i$, interpreted as the additive noise variable, is assumed to be independent of variables in $X_{\mathrm{PA}_i}$ and mutually independent with variables in $\mathcal{Z} \setminus Z_i$.

Rolland et al. (2022) proposed an order-based algorithm for this model, further assuming that $f_i$ is a twice-differentiable nonlinear function and $Z_i$ is Gaussian noise. This method enables the identification of leaf nodes based on the diagonal of the Hessian of the log-likelihood. Before proceeding to the method for identifying leaves, we first derive an analytical expression for the score following Lemma 2 in Rolland et al. (2022). The score is written as follows:

$$
\begin{aligned}
\nabla_{x_j} \log p(\mathbf{x}) &= \nabla_{x_j} \log \prod_{i=1}^{d} p(x_i \mid x_{\mathrm{PA}_i}) \\
&= \nabla_{x_j} \sum_{i=1}^{d} \log p(x_i \mid x_{\mathrm{PA}_i}) \\
&= \nabla_{x_j} \sum_{i=1}^{d} \log p(x_i - f_i) \quad \text{(where } z_i = x_i - f_i(x_{\mathrm{PA}_i})) \\
&= \frac{\partial \log p(x_j - f_j(x_{\mathrm{PA}_j}))}{\partial x_j} - \sum_{i \in \mathrm{CH}_j} \frac{\partial f_i}{\partial x_j} \frac{\partial \log p(x_i - f_i(x_{\mathrm{PA}_i}))}{\partial x}.
\end{aligned}
$$

where $\mathrm{CH}_j$ represents the children of the variable $j$. As a result, $\frac{\partial}{\partial x_j} \nabla_{x_j} \log p(\mathbf{x}) = a$, where $a$ is a constant, Consequently, the variance of the diagonal elements of the Hessian is zero (i.e. $\mathrm{Var}_{\mathbf{X}}[H_{j,j}(\log p(\mathbf{x}))] = 0$) if and only if node $j$ is a leaf node.

However, existing computational methods for calculating the Hessian struggle to scale efficiently as the number of variables and samples increases, limiting the scalability of Rolland et al. (2022). To address this, Sanchez et al. (2022) introduced a diffusion-based model that efficiently computes the Hessian, enabling the method to scale to larger datasets, both in terms of sample size and number of variables, while maintaining comparable performance to Rolland et al. (2022).

**Causal Discovery with Incomplete Data.** Several extensions of the PC algorithm have been developed to learn causal graphs from incomplete data (Tu et al., 2019; Gain & Shpitser, 2018), utilizing only the fully observed samples while mitigating biases in conditional independence tests. Another prominent family of methods relies on Expectation-Maximization (Dempster et al., 1977), where missing values are iteratively inferred while simultaneously learning the causal structure. Building on the continuous optimization techniques introduced by NOTEARS (Zheng et al., 2018), MissDAG (Gao et al., 2022) extends this approach to continuous identifiable Additive Noise Models (ANMs), using approximate posterior inference via Monte Carlo and rejection sampling when the exact posterior is unavailable. MissOTM (Vo et al., 2024) introduces a score-based method that leverages optimal transport to learn causal structures from incomplete data. A distinct approach is taken by VISL (Morales-Alvarez et al., 2022), which employs amortized variational inference in a Bayesian framework. Unlike MissOTM and MissDAG, VISL assumes a latent low-dimensional factor that captures the essential structure of the data based on observed variables. The latent factors are then used to reconstruct the complete data and discover the underlying causal graph.

## D.2. Evaluation Metrics

For each method, we compute the

**SHD.** Structural Hamming distance between the output and the true causal graph, which counts the number of missing, falsely detected, or reversed edges.

**Order Divergence.** (Rolland et al., 2022) propose this quantity for measuring how well the topological order is estimated. For an ordering $\pi$, and a target adjacency matrix A, we define the topological order divergence $D_{\text{top}}(\pi, \mathbf{A})$ as

$$D_{\text{top}}(\pi, \mathbf{A}) = \sum_{i=1}^{d} \sum_{j:\pi_i > \pi_j} \mathbf{A}_{ij} \tag{18}$$

## D.3. Model Architecture

We apply a perturbation of $\sigma = 0.1$ and jointly optimize Eq. (7) and Eq. (8). The model is a simple MLP with 5 Linear layers, LeakyReLU activation, Layer Normalization, and a Dropout rate of 0.2 in the first layer. The learning rate is set to 0.001. The first two layers have a latent size of $\max(128, 3 \times d)$, while the last three use a latent size of $\max(1024, 5 \times d)$. Training is conducted with a batch size of 128 for 150 epochs. The time efficiency is shown in the figure 4 with ER graph model across various dimensions. All experiments are executed on an Intel(R) Xeon(R) Gold 6448H CPU. The algorithm is presented in Algorithm D.3.

---

**Algorithm 3** MissScore for Causal Discovery with Missing Data

---

1: **Input:** Observed data $\mathbf{x}_{\text{obs}}$; score models $\mathbf{s}_1(\cdot; \boldsymbol{\theta})$, $\mathbf{s}_2(\cdot; \boldsymbol{\theta})$
2: **Initialize:** $\pi = []$; nodes $= \{1, \ldots, d\}$
3: $n, d \leftarrow$ shape of $\mathbf{x}_{\text{obs}}$
4: **for** $k = 1$ to $d$ **do**
5:     Jointly train the score models $\mathbf{s}_1(\boldsymbol{\theta})$ and $\mathbf{s}_2(\boldsymbol{\theta})$ using $\mathbf{x}_{\text{obs}}$ with Algorithm 1
6:     Generate $n$ samples $\tilde{\mathbf{x}}_{\text{new}}$ using Algorithm 2 with bootstrapping
7:     Estimate the second-order score $\mathbf{s}_2(\tilde{\mathbf{x}}_{\text{new}})$ using $\mathbf{s}_2(\boldsymbol{\theta})$
8:     $V_j = \text{Var}_X[\text{diag}(\mathbf{s}_2(\tilde{\mathbf{x}}_{\text{new}}))]$
9:     $\ell \leftarrow \arg\min_{j \in \text{nodes}} V_j$    The leaf node
10:     $\pi \leftarrow [\ell, \pi]$ Update topological order
11:     nodes $\leftarrow$ nodes $- \{\ell\}$ Remove node $\ell$
12:     Remove the $\ell$-th column from $\mathbf{x}_{\text{obs}}$
13: **end for**
14: Obtain the final DAG using CAM pruning associated with the topological order $\pi$.

---

## D.4. Experimental Results

To evaluate MissScore in the context of causal discovery under missing data, we compare it with several state-of-the-art methods. These include MissDAG (Gao et al., 2022) and MissOTM (Vo et al., 2024), which are specifically designed for causal structure learning with incomplete data. In contrast, we also include two imputation-based pipelines that rely on MissForest (Stekhoven & Bühlmann, 2012) to complete the data prior to structure learning: DiffAN (Sanchez et al., 2022) (denoted as MissDiffAN), a diffusion-based score matching approach, and DAGMA (Bello et al., 2022) (denoted as MissForest), a continuous optimization method. Together, these baselines represent a broad range of causal discovery strategies, including continuous optimization, optimal transport, score-based methods, and imputation-then-learn pipelines.

We begin by conducting experiments using complete data, evaluating our approach alongside various missing mechanisms and different missing ratios. Additionally, we include order divergence in the Table. Our findings reveal that, with complete data, MissScore performs on par with DiffAN, as illustrated in Figure 6. Notably, among all settings, MissScore achieves performance similar to current state-of-the-art approaches while offering superior computational efficiency in terms of memory and time. This scalability is a key advantage where other methods may struggle.

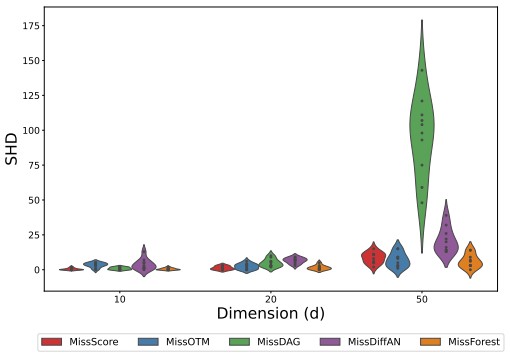 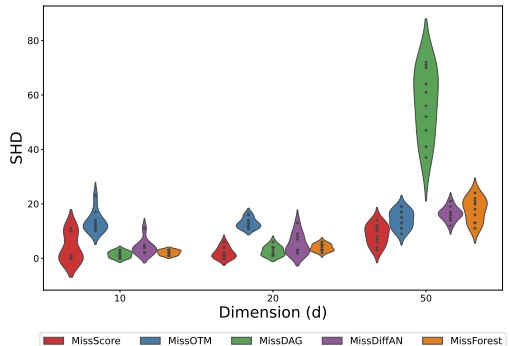

*Figure 6.* The data is generated using an ER graph model with different dimensions $d = \{10, 20, 50\}$ and an equal number of edges. Each dataset consists of 1000 samples. Left: $f_i$ is an MLP; Right: $f_i$ corresponds to MIM.

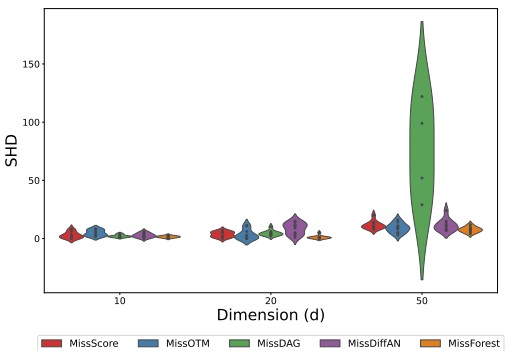 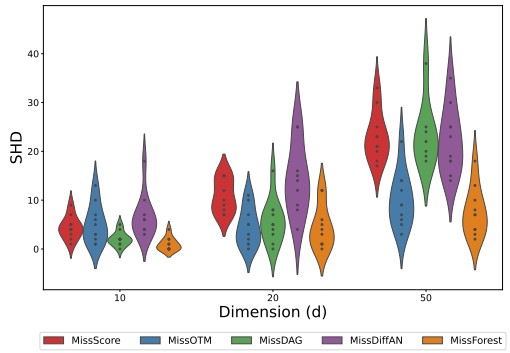

*Figure 7.* The data is generated under MCAR with missing ratios of 0.1 and 0.3, using an ER graph model with different dimensions $d = \{10, 20, 50\}$ and an equal number of edges. Each dataset consists of 1000 samples. $f_i$ corresponds to MLP. Left: SHD with missing ratio 0.1; Right: SHD with missing ratio 0.3.

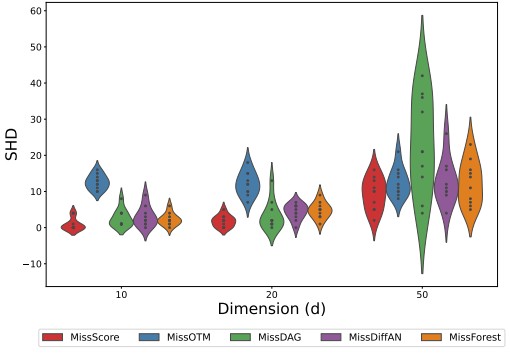 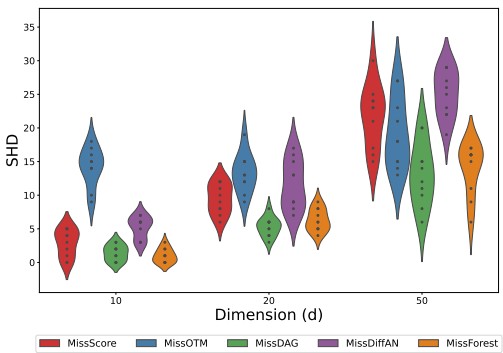

*Figure 8.* The data is generated under MCAR with missing ratios of 0.1 and 0.3, using an ER graph model with different dimensions $d = \{10, 20, 50\}$ and an equal number of edges. Each dataset consists of 1000 samples. $f_i$ corresponds to MIM. Left: SHD with missing ratio 0.1; Right: SHD with missing ratio 0.3.

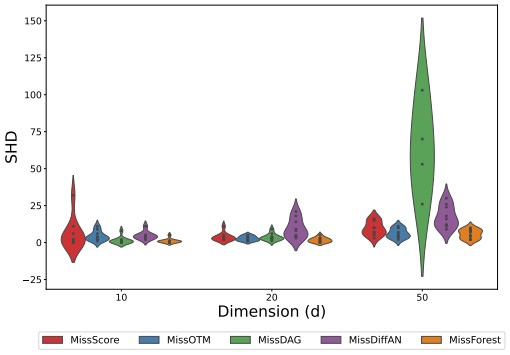 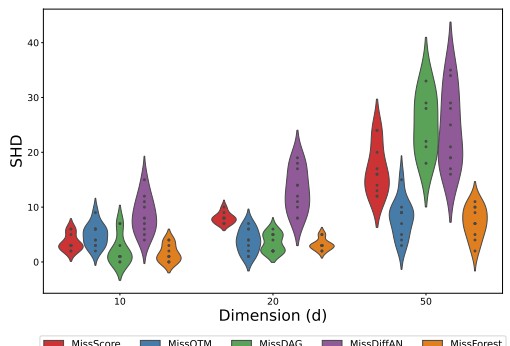

*Figure 9.* The data is generated under MAR with missing ratios of $0.1$ and $0.3$, using an ER graph model with different dimensions $d = \{10, 20, 50\}$ and an equal number of edges. Each dataset consists of 1000 samples. $f_i$ corresponds to MLP. Left: SHD with missing ratio 0.1; Right: SHD with missing ratio 0.3.

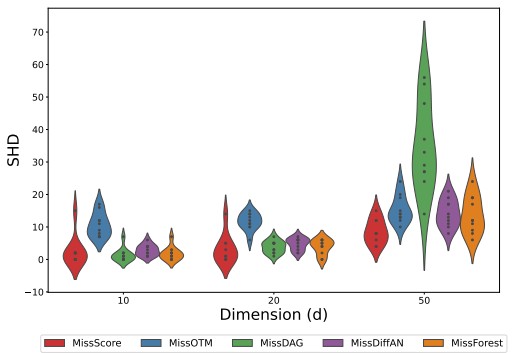 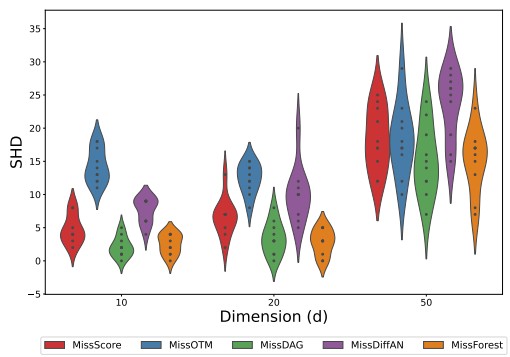

*Figure 10.* The data is generated under MAR with missing ratios of $0.1$ and $0.3$, using an ER graph model with different dimensions $d = \{10, 20, 50\}$ and an equal number of edges. Each dataset consists of 1000 samples. $f_i$ corresponds to MIM. Left: SHD with missing ratio 0.1; Right: SHD with missing ratio 0.3.

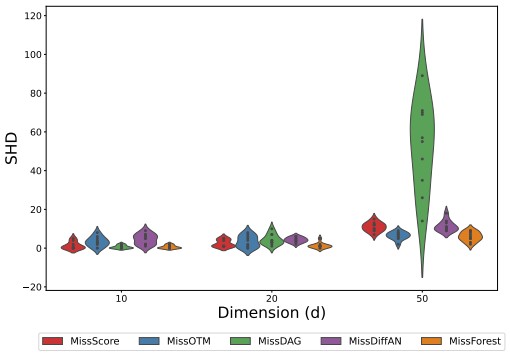 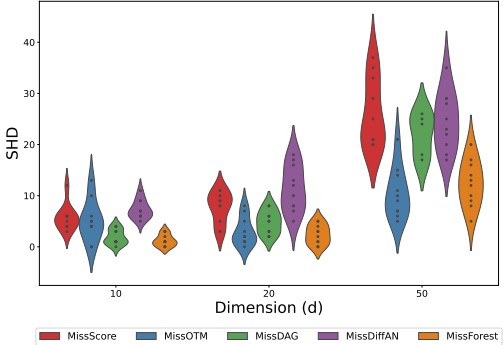

*Figure 11.* The data is generated under MNAR with missing ratios of $0.1$ and $0.3$, using an ER graph model with different dimensions $d = \{10, 20, 50\}$ and an equal number of edges. Each dataset consists of 1000 samples. $f_i$ corresponds to MLP. Left: SHD with missing ratio 0.1; Right: SHD with missing ratio 0.3.

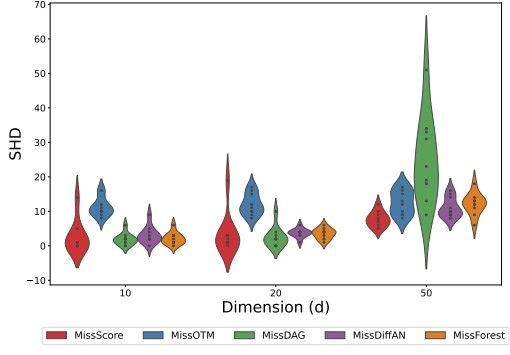 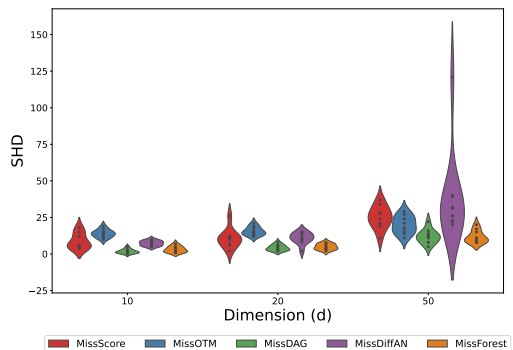

*Figure 12.* The data is generated under MNAR with missing ratios of 0.1 and 0.3, using an ER graph model with different dimensions $d = \{10, 20, 50\}$ and an equal number of edges. Each dataset consists of 1000 samples. $f_i$ corresponds to MIM. Left: SHD with missing ratio 0.1; Right: SHD with missing ratio 0.3.

*Table 9.* Order divergence with missing ratios of $\alpha = \{0.1, 0.3\}$ across different missing data mechanisms. The ER graph model is considered with varying dimensions $d = \{10, 20, 50\}$, and an equal number of edges. Each dataset consists of 1000 samples and $f_i$ corresponds to MLP. Lower order divergence indicating better performance.

| Dimensions | Methods | MCAR | | MAR | | MNAR | |
|---|---|---|---|---|---|---|---|
| | | $\alpha = 0.1$ | $\alpha = 0.3$ | $\alpha = 0.1$ | $\alpha = 0.3$ | $\alpha = 0.1$ | $\alpha = 0.3$ |
| d=10 | MissScore | $1.62 \pm 0.99$ | $1.60 \pm 0.66$ | $1.33 \pm 0.47$ | $1.89 \pm 0.75$ | $1.33 \pm 0.47$ | $1.44 \pm 0.50$ |
| | MissDiffAN | $2.00 \pm 1.18$ | $2.90 \pm 1.87$ | $2.75 \pm 1.30$ | $3.60 \pm 1.28$ | $2.60 \pm 1.20$ | $2.30 \pm 1.35$ |
| d=20 | MissScore | $2.10 \pm 0.94$ | $2.67 \pm 2.00$ | $2.70 \pm 1.85$ | $1.94 \pm 0.29$ | $1.70 \pm 1.78$ | $0.70 \pm 0.46$ |
| | MissDiffAN | $4.30 \pm 2.00$ | $4.22 \pm 2.35$ | $4.33 \pm 2.11$ | $3.00 \pm 1.00$ | $2.00 \pm 0.89$ | $3.40 \pm 1.20$ |
| d=50 | MissScore | $3.40 \pm 2.33$ | $4.00 \pm 3.10$ | $3.10 \pm 1.70$ | $4.60 \pm 2.91$ | $2.80 \pm 0.98$ | $4.10 \pm 3.30$ |
| | MissDiffAN | $4.50 \pm 2.91$ | $3.60 \pm 3.32$ | $8.33 \pm 3.65$ | $7.40 \pm 2.84$ | $3.50 \pm 1.75$ | $3.20 \pm 3.28$ |

*Table 10.* Order divergence with missing ratios of $\alpha = \{0.1, 0.3\}$ across different missing data mechanisms. The ER graph model is considered with varying dimensions $d = \{10, 20, 50\}$, and an equal number of edges. Each dataset consists of 1000 samples and $f_i$ corresponds to MIM. Lower order divergence indicating better performance.

| Dimensions | Methods | MCAR | | MAR | | MNAR | |
|---|---|---|---|---|---|---|---|
| | | $\alpha = 0.1$ | $\alpha = 0.3$ | $\alpha = 0.1$ | $\alpha = 0.3$ | $\alpha = 0.1$ | $\alpha = 0.3$ |
| d=10 | MissScore | $1.60 \pm 1.20$ | $1.20 \pm 0.60$ | $1.35 \pm 0.47$ | $1.88 \pm 0.93$ | $1.62 \pm 0.99$ | $1.38 \pm 0.48$ |
| | MissDiffAN | $2.22 \pm 2.10$ | $1.50 \pm 0.67$ | $2.50 \pm 1.36$ | $3.70 \pm 1.10$ | $2.40 \pm 2.20$ | $2.20 \pm 1.40$ |
| d=20 | MissScore | $1.70 \pm 0.90$ | $2.20 \pm 1.33$ | $1.82 \pm 1.29$ | $1.90 \pm 1.30$ | $1.56 \pm 0.68$ | $0.56 \pm 0.83$ |
| | MissDiffAN | $2.70 \pm 1.49$ | $3.8 \pm 1.47$ | $3.70 \pm 1.35$ | $3.10 \pm 2.21$ | $2.44 \pm 1.07$ | $2.70 \pm 1.68$ |
| d=50 | MissScore | $4.90 \pm 2.12$ | $4.00 \pm 1.41$ | $4.56 \pm 2.27$ | $4.60 \pm 2.24$ | $4.20 \pm 1.66$ | $4.40 \pm 1.85$ |
| | MissDiffAN | $5.56 \pm 2.45$ | $4.40 \pm 1.56$ | $7.90 \pm 2.39$ | $7.50 \pm 2.69$ | $5.00 \pm 1.61$ | $6.20 \pm 2.68$ |

