# OpenReview forum: "MissScore: High-Order Score Estimation in the Presence of Missing Data"
_ICML.cc/2025/Conference — ICML 2025 poster_

### Official Review · Reviewer_bpYG · 2025-03-07

**Overall Recommendation:** 3

**Summary:**

The paper introduces MissScore, a novel framework for high-order score estimation in the presence of missing data. Existing high-order score-based generative models assume data completeness, necessitating imputation when data is missing. MissScore extends Denoising Score Matching (DSM) to estimate high-order scores directly, eliminating the need for imputation and addressing a critical gap in the field. The authors establish strong theoretical foundations and derive objective functions tailored to different missing data mechanisms (MCAR, MAR). While the empirical evaluation is somewhat limited in the number of baselines and datasets, the results evidence MissScore’s effectiveness in sampling, data generation, and causal discovery, outperforming existing methods. This work represents a significant theoretical and practical contribution to score-based modeling.

**Claims And Evidence:**

The paper presents strong theoretical and empirical claims regarding the feasibility and effectiveness of high-order score estimation with missing data. It establishes a rigorous mathematical foundation by deriving objective functions for high-order score matching under MCAR and MAR mechanisms, demonstrating that high-order scores can be estimated without imputation.

Empirically, the authors validate MissScore primarily on synthetic datasets, with only one real-world dataset for synthetic data generation. This is somewhat limited, as generative modeling with missing data is typically evaluated across multiple datasets (e.g., UCI). Additionally, key references and comparisons to deep generative models for missing data are missing, making it difficult to fully assess MissScore’s relative performance. Furthermore, while the paper claims high-dimensional scalability, the use of a 100-dimensional Gaussian dataset may be overly simplistic and insufficient to support this claim.

Overall, expanding the empirical validation to more diverse real-world benchmarks and recent related work would strengthen the evidence.

The empirical analysis for causal discovery, however, appears reasonable.

**Essential References Not Discussed:**

VAEs for missing data have been quite successful in the recent years, and they are briefly commented in the Introduction section as methods that require training additional networks, which is not fully grounded. Many recent work are ignored for assessing the empirical Below I include some unreferenced VAE-based and diffusion-based related works.


### References

[1] Nazabal, Alfredo, et al. "Handling incomplete heterogeneous data using vaes." Pattern Recognition 107 (2020): 107501.

[2] Mattei, Pierre-Alexandre, and Jes Frellsen. "MIWAE: Deep generative modelling and imputation of incomplete data sets." International conference on machine learning. PMLR, 2019.

[3] Ma, Chao, et al. "VAEM: a deep generative model for heterogeneous mixed type data." Advances in Neural Information Processing Systems 33 (2020): 11237-11247.

[4] Peis, Ignacio, Chao Ma, and José Miguel Hernández-Lobato. "Missing data imputation and acquisition with deep hierarchical models and hamiltonian monte carlo." Advances in Neural Information Processing Systems 35 (2022): 35839-35851.

[5] Zhang, Hengrui, Liancheng Fang, and S. Yu Philip. "Unleashing the Potential of Diffusion Models for Incomplete Data Imputation." CoRR (2024).

[6] Chen, Zhichao, et al. "Rethinking the diffusion models for missing data imputation: A gradient flow perspective." Advances in Neural Information Processing Systems 37 (2024): 112050-112103.

[7] Zheng, Shuhan, and Nontawat Charoenphakdee. "Diffusion models for missing value imputation in tabular data." NeurIPS 2022 First Table Representation Workshop.

**Experimental Designs Or Analyses:**

The experimental design is well-structured and appropriate for the problem setting, but broader empirical validation would provide stronger support for the method’s real-world applicability.

**Methods And Evaluation Criteria:**

For synthetic data generation, the authors assess MissScore’s effectiveness in estimating high-order scores and improving sampling efficiency. They compare Langevin dynamics and Ozaki sampling to demonstrate the advantages of incorporating second-order score information. However, as noted previously, only one real-world dataset is used in this setting, which limits the generalizability of the results. Expanding the evaluation to multiple real-world datasets, as is common in generative modeling research, would strengthen the empirical validation.

For causal discovery, the authors evaluate MissScore’s ability to recover causal structures from incomplete data. The method is compared against baselines such as MissForest + DiffAN/DAGMA and MissDAG/MissOTM, using Structural Hamming Distance (SHD) and Order Divergence as evaluation metrics. This part of the evaluation appears more comprehensive and methodologically sound.

Overall, the evaluation methodology is valid and well-structured, but its impact is somewhat constrained by the limited number of real-world datasets used for validation.

**Other Comments Or Suggestions:**

N/A

**Other Strengths And Weaknesses:**

All strengths and weaknesses are already commented above.

**Questions For Authors:**

I do not have any further questions.

**Relation To Broader Scientific Literature:**

The paper builds on foundational work in score estimation and extends recent advances in high-order score modeling to address missing data. It differentiates itself from imputation-based approaches (e.g., VAEs, GANs) and diffusion-based methods like MissDiff by directly estimating high-order scores from incomplete data. The discussion on VAEs for missing data does not reflect the state of the art.

However, the paper lacks direct comparisons to several generative models designed for missing data, both in citations and empirical evaluation. Expanding the related work discussion and empirical benchmarking would better position MissScore within the broader literature.

Overall, the paper makes a significant theoretical contribution, but a more comprehensive comparison to existing deep generative models would strengthen its impact.

**Theoretical Claims:**

The theoretical contributions are novel, well-structured, and mathematically sound, making this one of the strongest aspects of the paper. The results significantly advance the understanding of high-order score estimation in settings with missing data, and lay the groundwork for further exploration in score-based modeling or more complex missing patters.

---

> ### Author Rebuttal · Authors · 2025-04-01
>
> # To Reviwer bpYG
>
> We sincerely thank you for your valuable feedback and the time spent evaluating our work! Below, we summarize the main concerns and provide detailed responses to each point.
>
> ---
>
> ### **Related work on VAE- and diffusion-based methods.**
>
> Thank you for the suggestions. We agree that these works are highly relevant and will update the related work section to include the provided references and a recent diffusion-based method [8]. The revised paragraph is:
>
> >Deep generative models have emerged as a powerful framework for learning from incomplete data. VAEs are widely used for handling incomplete data due to their flexibility in modeling complex distributions. MIWAE [2] extends the importance-weighted autoencoder to handle missing-at-random (MAR) data, yielding theoretically grounded single and multiple imputations. HI-VAE [1] addresses heterogeneous variables by assigning each data type a tailored likelihood, with a shared variational mechanism that balances learning across different attributes. VAEM [3] similarly supports mixed-type data using a two-stage process—first learning per-feature marginal VAEs, then a dependency VAE to model correlations among the latent codes—while hierarchical variants [4] further increase representational capacity at the cost of more challenging inference. Diffusion models have also gained traction for missing data imputation. TabCSDI [7] adapts conditional score-based diffusion to tabular data, addressing categorical variables and modeling the conditional distribution of missing features. NewImp [6] reinterprets diffusion-based imputation as a gradient flow in probability space, offering theoretical insights, while DIFFPUTER [5] combines the EM algorithm with diffusion models in an iterative framework. ForestDiffusion [8] enhances interpretability and efficiency by replacing neural score estimators with gradient-boosted trees, enabling direct learning from partially observed tabular data. While these methods can be used for imputation along with generation, our approach takes a different direction by directly estimating high-order score functions from incomplete data. This enables downstream tasks such as sampling and causal discovery without explicit imputation.
>
> ---
>
> ### **Broader Empirical Validation and Additional Comparisons.**
>
> We evaluate our method on 27 real-world classification and regression datasets for the task of generation, sourced from the UCI Machine Learning Repository and scikit-learn. Details of the datasets, evaluation metrics, and more comparison methods are provided in Appendices B.1, B.2, and B.3 in ForestDiffusion [2]. This comprehensive evaluation benchmarks MissScore against established methods across diverse datasets, ensuring robustness and reliability.
>
> | Method         | W_train ↓ | W_test ↓ | cov_train ↑ | cov_test ↑ | R²_fake ↑ | F1_fake ↑ | F1_disc ↓ | P_bias ↓ | cov_rate ↑ |
> |----------------|--------|--------|----------|----------|--------|--------|--------|--------|---------|
> | Stasy          | 3.40   | 3.67   | 0.38     | 0.63     | 0.27   | 0.64   | 0.46   | **1.09** | 0.36    |
> | TabDDPM        | 4.36   | 4.80   | **0.72** | 0.71     | **0.58** | 0.61 | **0.42** | 1.16   | **0.56** |
> | Forest-VP      | 1.84   | 2.14   | 0.53     | 0.78     | 0.53   | **0.71** | **0.42** | 1.16   | 0.43    |
> | Forest-Flow    | 1.82   | 2.12   | 0.67     | **0.84** | 0.55   | 0.69   | 0.43   | 1.16   | 0.50    |
> | MissScore-Langevin | 1.85   | 2.15   | 0.65     | 0.82     | 0.54   | 0.68   | 0.44   | 1.18   | 0.49    |
> | MissScore-Ozaki    | **1.79** | **2.11** | 0.70 | 0.83     | **0.58** | 0.70 | 0.43   | 1.15   | 0.52    |
>
> Table 2. Tabular data generation with incomplete data (27 datasets, 3 experiments per dataset, 20% missing values); results show the mean.
>
> ---
>
> ### **Insufficient Support for Scalability Claims.**
>
> Thank you for highlighting this point. We agree that the 100-dimensional Gaussian dataset alone may be too simplistic to substantiate a strong claim of high-dimensional scalability. In the causal discovery setting (Section 6), MissScore is evaluated on datasets with up to 50 variables—a scale typically regarded as high-dimensional in this context. We also reported results on 27 real-world tabular datasets, with dimensionalities reaching up to 90. While more extreme high-dimensional scenarios remain unexplored, we will revise the manuscript to more accurately reflect the scalability explored in our current evaluation.
>
> ---
>
> Thank you for reading our rebuttal! We hope the points discussed above have fully addressed your concerns. Please let us know if any questions remain or if further clarification is needed. We sincerely appreciate your time and thoughtful evaluation!
>
> ---
>
> ### **References**
>
> Refs [1]-[7] are the same as the list you provided. Apologies for the brevity due to word limits.
>
> [8] Jolicoeur-Martineau et al. (2024). *Generating and imputing tabular data via diffusion and flow-based gradient-boosted trees.*

---

> > ### Comment · Reviewer_bpYG · 2025-04-09
> >
> > Thank you for addressing my concerns. The additional empirical evaluation significantly strengthens the paper, and I will recommend acceptance.
> >
> > I also wanted to mention that I mistakenly posted the following as an “Official Comment,” not realizing it wouldn’t be visible to the authors. Given the limited time remaining in the rebuttal period, I understand that a response is unlikely at this point. However, I’m reposting it here in case it can serve as constructive feedback moving forward:
> >
> > > “I find the analysis of the UCI benchmark results in Table 2 to be lacking. Based on the reported metrics, the proposed method appears competitive, though it does not consistently outperform the baselines. Could the authors provide further analysis or insight into these results? In particular, it would be helpful to understand the scenarios in which MissScore is especially beneficial, and how it compares qualitatively or in practical applications, despite the mixed quantitative performance.”
> >
> > Thank you again for the thoughtful rebuttal and for strengthening the submission.

---

> > > ### Author Response · Authors · 2025-04-09
> > >
> > > **Response to: “It would be helpful to understand the scenarios in which MissScore is especially beneficial, and how it compares qualitatively or in practical applications, despite the mixed quantitative performance.”**
> > >
> > > Thank you for the thoughtful feedback. While MissScore-Ozaki may not always outperform all baselines across every metric, we have found it to be beneficial in several practical scenarios:
> > >
> > > - Scenarios where MissScore performs well include:
> > >   - *Datasets with reasonably separable class structures and moderate sample sizes* (e.g., *seeds*), where score-based modeling ensures high sample quality and supports effective downstream classification.
> > >   - *Datasets with complex but moderately balanced class distributions* (e.g., *parkinsons*), where MissScore can model the nuanced relationships between classes.
> > >   - *Moderate to high-dimensional datasets* (e.g., *qsar biodegradation*), where second-order score matching effectively captures fine-grained variations and non-linear dependencies within the feature space.
> > >
> > > In these scenarios, MissScore-Ozaki typically achieves lower Wasserstein distances and higher R²_fake and F1_fake scores, indicating that the generated data closely approximates the real distribution while also performing well on downstream tasks.
> > >
> > > - More challenging scenarios include:
> > >   - *Extremely small datasets* (e.g., *concrete slump*, with only 103 samples), where the flexibility of the score network may lead to instability in higher-order score matching.
> > >   - *Datasets with many classes and relatively few samples per class* (e.g., *libras*, with 15 classes and 24 samples each), where training the score estimator becomes more difficult and variance increases during generation.
> > >
> > > Due to time limitations, we were only able to analyze a subset of the datasets. As such, we will expand the analysis to include additional datasets in the revision. Thank you again for your constructive feedback as we move forward with this work.

---

### Official Review · Reviewer_Bf4B · 2025-03-13

**Overall Recommendation:** 4

**Summary:**

They introduce a method for estimation high-order scores in the presence of missing data with either MAR or MCAR. For MAR, they need to make use of logistic regression to estimate a needed quantity. They propose stability improvements with variance reductions.

They show low error in the estimation of the first and second-order score with extra missing data.

**Claims And Evidence:**

- novel score-based framework for learning high-order scores in the presence of missing data (true when there is missing data)
- show efficiently and accurately approximates high-order scores with missing data (true, good evidence)
- improves both sampling speed and data quality in data generation tasks, with the quality of the generated samples validated across several downstream tasks (Ozaki Sampling does converge in fewer steps than Langevin, but they only show this a 2D toy setting; for quality, the evidence is weak, they need more experiments on different benchmarks)

To add more details on the faster speed, this seems wrong since Ozaki is probably slower in actually wallclock time. Please add wallclock or FLOPS or similar metrics to actually assess speed, not just in number of steps. Since second-order steps are more expensive.

**Essential References Not Discussed:**

No

**Experimental Designs Or Analyses:**

"Methods And Evaluation Criteria"

**Methods And Evaluation Criteria:**

For tabular data: Fidelity and utility are okay simple metrics, but they are limited.
- SDmetrics is mostly 1D and some 2D metrics, its limited.
- Utility is not the correct name, the Stasy people branded it this way, but this metrics already exist with different name. Its called machine learning (ML) efficiency, see https://arxiv.org/pdf/2209.15421. Please rectify.

You should consider metrics such as Wasserstein distance, see https://arxiv.org/abs/2309.09968 for a very strong benchmark with many metrics. I recommend using a wide benchmark like this containing many metrics with many datasets, right now only using the Census dataset is not enough. One dataset is very poor evidence because it could be fished for the one that shows the best result on your method.

Only using swiss roll to say it converge in fewer steps, is not enough.

The causal discovery experiments that uses MissForest as imputation for methods that don't handle missing data are good.

**Other Comments Or Suggestions:**

- please rasterize the figures of swiss rolls, there is too many points, its make the pdf unresponsive.

**Other Strengths And Weaknesses:**

The main issue right now is the experiments, if the authors can improve the depth of their experiments for both speed and quality (see my suggestions above) it would warrant increasing my score.

I am a bit disappointing that the paper only focus on Langevin versus Ozaki sampling when they could derive a second-order diffusion or something like this. This would make the paper stronger for sure.

**Questions For Authors:**

.

**Relation To Broader Scientific Literature:**

They take existing score estimation and adapt it for when there is missing data (theory and loss function), which is very useful.

**Theoretical Claims:**

I haven't checked the details, but their error rate experiments seems to suggests that their derivations are correct given the low errors.

---

> ### Author Rebuttal · Authors · 2025-04-01
>
> # To Reviewer Bf4B
>
> We appreciate the reviewer’s thoughtful comments and the time spent on reviewing our paper! Below, we summarize the concerns and suggestions and provide detailed responses to each.
>
> ---
>
> ### **Faster Convergence Rate, Not Faster Wall-Clock Time.**
>
> Ozaki sampling achieves **faster convergence rate** in terms of **iteration count**, as supported by theoretical results [1] under certain conditions and validated by our experiments (e.g., Figure 2). However, we acknowledge that this does not necessarily correspond to faster wall-clock time due to the additional computational overhead. To assess this, we measured the average wall-clock training time per iteration and the results are summarized below:
>
> | **Sampling Methods** | **Time (s)** |
> |---------------------|--------------|
> | Langevin | 0.00037 ± 0.00017 |
> | Ozaki | 0.00043 ± 0.00053 |
>
> As expected, Ozaki sampling incurs a longer wall-clock time compared to Langevin. In our experiments, we consider Ozaki sampling with $s_2$ using only its diagonal, as described in Appendix C. This diagonal approximation reduces the computational overhead, making Ozaki sampling comparable to Langevin in terms of wall-clock time per iteration while retaining its advantage of faster convergence. We will clarify this in the revision that our claims of faster convergence (or speed) refer strictly to iteration count, not wall-clock time. Thank you again for your valuable feedback!
>
> ---
>
> ### **Higher Quality of Ozaki Sampling.**
>
> Our wall-clock time analysis shows that Ozaki sampling incurs higher computational cost per iteration compared to Langevin. Despite this, Ozaki sampling often achieves better sample quality. Extensive experiments on 27 real-world datasets demonstrate that Ozaki sampling consistently produces higher-quality samples than Langevin for the same number of iterations. Even when we increased Langevin's sampling time to match that of Ozaki (MissScore-Langevin2), the improvements were modest, with Ozaki still outperforming the longer-running Langevin across all metrics — further emphasizing the advantage of incorporating second-order information.
>
> | Method | W_train ↓  | W_test ↓   | cov_train ↑ | cov_test ↑ | R²_fake ↑  | F1_fake ↑   | F1_disc ↓   | P_bias ↓   | cov_rate ↑  |
> |-----------------------|------------|------------|-------------|------------|------------|-------------|-------------|------------|-------------|
> | MissScore-Langevin|1.85(0.52)|2.15(0.57)|0.65(0.04)|0.82(0.03)|0.54(0.11)|0.68(0.04)|0.44(0.02)|1.18(0.33)| 0.49(0.11)|
> | MissScore-Langevin2|1.83(0.50)|2.13(0.55)|0.67(0.04)|0.83(0.05)|0.55(0.11)|0.69(0.05)|0.44(0.01)|1.19(0.34)| 0.50(0.11)|
> | MissScore-Ozaki|1.79(0.50)|2.11(0.55)|0.70(0.03)|0.83(0.03) | 0.58(0.10) | 0.70(0.04) | 0.43(0.02)|1.15(0.30)|0.52(0.10)|
>
> *Table 1: Tabular data generation with incomplete data across 27 datasets (3 experiments per dataset, 20% missing values). MissScore is evaluated using Langevin and Ozaki with the same number of iterations, and Langevin2 with increased sampling iterations (same sampling time as Ozaki); results show the mean (standard error).*
>
> ---
>
> ### **A broader benchmark incorporating multiple datasets and advanced metrics.**
>
> We kindly refer to the **Broader Empirical Validation and Additional Comparisons** section in our response to **Reviewer [bpYG]**, where we address similar concerns.
>
> ---
>
> ### **Beyond Langevin vs. Ozaki: Broader Applicability of Second-Order Information with Missing Data.**
>
> Thank you for your comment. We would like to clarify that:
> - Our primary goal is to address the challenge of missing data through the development of a high-order score estimation framework under partial observability, which avoids the need for full data or expensive automatic differentiation.
> - This framework enables downstream tasks such as causal discovery, which requires second-order information for capturing complex structural dependencies.
> - While our method also leads to better sampling performance (e.g., Ozaki), this is one of the benefits brought by second-order estimation, and not the primary focus of our work.
>
> ---
>
> ### **MissForest vs. MissScore.**
>
> MissForest captures dependencies through iterative imputation, making it a strong baseline for causal discovery with missing data. However, it becomes computationally expensive as dimensionality increases (as shown in Fig. 4b). In contrast, our method achieves comparable performance in simulations while being more efficient. Our new experiment on the Sach dataset (response to Reviewer [ugL7]), shows that MissScore maintains stable performance even with more complex dependencies, outperforming MissForest in this regard.
>
> ---
>
> ### **Other Suggestions.**
> We will update the manuscript to use "Machine Learning (ML) efficiency" and rasterize the Swiss roll figures to resolve the PDF rendering issue.
>
> ---
>
> ### **Ref.**
>
> [1] Dalalyan et al. (2017). *User-friendly guarantees for the Langevin Monte Carlo with inaccurate gradient.*

---

> > ### Comment · Reviewer_Bf4B · 2025-04-01
> >
> > Thank you for addressing most of my concerns. I am updating my score from 2 to 4.

---

> > > ### Author Response · Authors · 2025-04-01
> > >
> > > We sincerely appreciate your thoughtful evaluation and updated score! Your constructive feedback has been invaluable in guiding our revisions, and we are pleased that our efforts have addressed your concerns. Thank you for taking the time to review our work and for helping us enhance its quality!

---

### Official Review · Reviewer_ugL7 · 2025-03-15

**Overall Recommendation:** 3

**Summary:**

The authors propose a higher-order score estimation method for missing data. They first introduce the correct learning objectives in Theorems 3.3 to 3.5 and then formulate a multi-task objective to learn both the first and second-order scores with variance reduction to improve stability. The authors validate the effectiveness of the proposed method on a Swiss Roll dataset, a simulated Bayesian network dataset, and a Census dataset, comparing it against two baseline methods. The method successfully recovers the first and second-order oracle scores and generates high-quality samples. The authors further investigate the application of the proposed method for causal discovery and conduct experiments on an ER graph, comparing it against four baseline methods.

**Claims And Evidence:**

I believe the theorems are well proved.

**Essential References Not Discussed:**

The literature is discussed comprehensively.

**Experimental Designs Or Analyses:**

See weakness part below.

**Methods And Evaluation Criteria:**

Yes. MSE for whether the score function is learned, fidelity and utility for generation quality, and SHD for causal discovery.

**Other Comments Or Suggestions:**

From the background in Section 6, I understand that the proposed method can avoid the two-step estimation process, where data is first imputed and then used for causal discovery. However, it would be clearer if the authors could provide a more detailed explanation of the process for estimating the DAG and its relationship with ANM.

**Other Strengths And Weaknesses:**

Strengths: the topic is interesting and the derivation is innovative; the paper is well-written

Weakness: the experimental results are somehow weak to support the effectiveness of the proposed methods. All of the experiments are simulations or small datasets. It would be better for authors to conduct experiments on one real dataset for causal discovery and conduct more real experiments for score estimation.

**Questions For Authors:**

1. For Theorems 3.3–3.5, do the authors consider it difficult to generalize the conclusions to MNAR? If I understand correctly, all we do is incorporate $\mathbb{E} _{\tilde{x} | x, m}$ into the least squares loss. If $\mathbb{E} _{\tilde{x} | x, m}[1 - m]$ is not zero, then the denoising score matching objective is also equivalent to the score matching objective.

2. I cannot find the discussion of the data generation process for the experiments used in Figure 1 and Table 1. I am wondering what the ground truth score used in this experiment is. How is the ground truth score for missing data defined?

**Relation To Broader Scientific Literature:**

I believe that studying higher-order score estimation for missing data could improve current methods for imputation and generation.

**Theoretical Claims:**

I check the proof in Appendix A.

---

> ### Author Rebuttal · Authors · 2025-04-01
>
> # To Reviewer ugL7
> We appreciate your thoughtful comments and the time spent on reviewing our paper. Below, we summarize the concerns and suggestions and provide detailed responses to each.
>
> ---
>
> ### **Experiment on Real Dataset for Causal Discovery.**
> The Sachs dataset [1] models a network of cellular signaling pathways with 11 continuous variables and 7,466 samples. It is a widely used benchmark for causal discovery, providing cell signaling data with established causal relationships. The following table presents the Structural Hamming Distance (SHD↓) on the Sachs dataset under MCAR missingness at varying rates (10%, 30%, and 50%):
>
> | Method         | 10%  | 30%  | 50% |
> |----------------|------|------|-----|
> | MissOTM        | 44   | 42   | 43  |
> | MissForest     | 22   | 26   | 26  |
> | MissDAG        | 23   | 21   | 20  |
> | MissDiffAN     | 24   | 23   | 21  |
> | **MissScore**  | 22   | 22   | 19  |
>
> ---
>
> ### **More Experiments on Real-world Datasets.**
> We kindly refer to the **Broader Empirical Validation and Additional Comparisons** section in our response to **Reviewer [bpYG]**, where we address similar concerns.
>
> ---
>
> ### **Whether the theoretical results in Theorems 3.3–3.5 can be extended to the MNAR setting.**
>
> We thank the reviewer for the insightful question! For Theorems 3.3-3.5, the equivalence between denoising score matching and true score matching holds under MCAR and MAR because missingness is **independent** of unobserved values. This critical independence allows us to estimate $𝔼_{m|x}[1-m]$ from observed data, which ensures unbiased score estimation through the DSM loss. However, under MNAR, the missingness **depends** on the unobserved values themselves, creating fundamental identifiability issues. Even when $\mathbb{E}_{\bar{x}|x, m}[1-m]$ is not zero, the DSM objective cannot be shown equivalent to the true score matching objective because the necessary quantities cannot be reliably estimated from observed data alone. Generalizing our theoretical results to MNAR therefore remains an open challenge. That said, we have also conducted experiments under MNAR and observed promising empirical results. We acknowledge this limitation in the conclusion and highlight it as an important direction for future work.
>
> ---
>
> ### **The data generation process and the definition of ground truth scores used in Fig.1 and Table.1.**
>
> The data generation process is detailed in Sections 4.1, 4.2, and Appendix B. We generate data from a known multivariate Gaussian distribution $\mathcal{N}(\mu,\Sigma)$, allowing the ground truth scores to be computed analytically as $s_1(x) =-\Sigma^{-1}(x-\mu)$ and $s_2(x)=-\Sigma^{-1}$. Partially observed data is then generated under an MCAR mechanism, and the predicted scores are compared to these analytical ground truth values during evaluation.
>
> ---
>
> ### **Explanation of the DAG estimation process and its connection to ANM.**
> Briefly, our method leverages a structural property of additive noise models (ANMs), where the diagonal of the log-density Hessian reveals information about causal structure [3]. Specifically, for a leaf node $x_j$, the diagonal Hessian term: $\frac{\partial^2 \log p(x)}{\partial x_j^2}$ has zero variance across the data distribution. MissScore leverages this by estimating second-order scores directly from incomplete data, avoiding imputation. The DAG proceeds in three steps:
>
> 1. Score Estimation: We train first- and second-order score models using our high-order DSM objective with missing data handling.
> 2. Node Ordering: At each iteration, the variable with the lowest variance in its diagonal Hessian across samples is selected as a leaf and added to the causal ordering.
> 3. DAG Construction: A full topological order is recovered and pruned using the CAM pruning [4] to infer edge directions and finalize the DAG.
>
> We will revise Section 6 to more clearly summarize the DAG estimation procedure and its connection to the ANM, while referring readers to Appendix D for full details, including the formal ANM formulation, the role of second-order scores in identifying leaf nodes, the complete causal discovery pipeline (Algorithm 3), evaluation metrics, and extensive results under various missingness mechanisms.
>
> ---
>
> Thank you for reading our rebuttal! We hope the responses provided have fully addressed your concerns. If any questions remain or further clarification is needed, please feel free to let us know. Thank you again for your time, thoughtful feedback, and valuable insights!
>
> ---
>
> ### **References**
>
> [1] Sachs et al. (2005). *Causal protein-signaling networks from single-cell data.*
>
> [2] Jolicoeur-Martineau et al. (2024). *Generating and imputing tabular data via diffusion and gradient-boosted trees.*
>
> [3] Rolland et al. (2022). *Score matching enables causal discovery of nonlinear additive noise models.*
>
> [4] Bühlmann et al. (2014). *CAM: Causal additive models, high-dimensional order search, and penalized regression.*

---

### Decision · Program_Chairs · 2025-05-01

**Decision:**

Accept (poster)

**Comment:**

The manuscript proposes a framework for high-order score estimation applied to data with missing values within score-based modelling. The reviewers found the manuscript well-written and well-structured, and the theoretical contributions novel and sound. However, the experimental results are weak since they are limited to mainly simulated data or small datasets. The authors addressed this during the rebuttal by adding evaluations of multiple UCI datasets (these results must be included in the final revision of the manuscript). The reviewers also pointed to a lack of methodical and empirical comparison to other generative models designed for missing data. During the rebuttal, the authors proposed a revised version of the related works section, but we still urge the authors to also include these methods in the empirical evaluation. Given that the authors addressed most of the reviewer's concerns during the rebuttal period, we recommend accepting the paper if there is room in the program.